## Registered report

psychology

replication, study selection, consensus

**Author for correspondence:**
Merle-Marie Pittelkow
e-mail: m.pittelkow@rug.nl

# The process of replication target selection in psychology: what to consider?

Merle-Marie Pittelkow[1], Sarahanne M. Field[2],
Peder M. Isager[3], Anna E. van't Veer[4],
Thomas Anderson[5], Scott N. Cole[6], Tomáš Dominik[7],
Roger Giner-Sorolla[8], Sebahat Gok[9], Tom Heyman[4],
Marc Jekel[10], Timothy J. Luke[11], David B. Mitchell[12],
Rik Peels[13], Rosina Pendrous[14,15], Samuel Sarrazin[16],
Jacob M. Schauer[17], Eva Specker[18], Ulrich S. Tran[18],
Marek A. Vranka[19], Jelte M. Wicherts[20],
Naoto Yoshimura[21,22], Rolf A. Zwaan[23] and
Don van Ravenzwaaij[1]

[1]Department of Psychometrics and Statistics, Rijksuniversiteit Groningen, Groningen, The Netherlands
[2]Centre of Science and Technology Studies, Leiden University, Leiden, the Netherlands
[3]Oslo New University College, Oslo, Norway
[4]Methodology and Statistics Unit, Institute of Psychology, Leiden University, Leiden, the Netherlands
[5]Department of Psychology, University of Toronto, Toronto, Canada
[6]School of Education, Language and Psychology, York St John University, York, UK
[7]Institute for Interdisciplinary Brain and Behavioral Sciences, Chapman University, Orange, CA, USA
[8]School of Psychology, University of Kent, Canterbury, UK
[9]Program in Cognitive Science, Department of Instructional Systems Technology, Indiana University, Bloomington, IN, USA
[10]Social Psychology, University of Cologne, Cologne, Germany
[11]Department of Psychology, University of Gothenburg, Gothenburg, Sweden
[12]WellStar College of Health and Human Services, Kennesaw State University, Kennesaw, GA, USA
[13]Philosophy Department and the Faculty of Religion and Theology, Vrije Universiteit, Amsterdam, The Netherlands
[14]Centre for Contextual Behavioural Science, School of Psychology, University of Chester, Chester, UK
[15]Institute of Applied Health Research, University of Birmingham, Birmingham, UK
[16]Maison de santé pluridisciplinaire Pasteur, Chevilly-Larue, France

[17]Department of Preventive Medicine - Division of Biostatistics, Feinberg School of Medicine, Northwestern University, Chicago, IL, USA
[18]Department of Cognition, Emotion, and Methods in Psychology, Faculty of Psychology, University of Vienna, Vienna, Austria
[19]Faculty of Social Sciences, Charles University, Prague, Czech Republic
[20]Department of Methodology and Statistics, Tilburg University, Tilburg, The Netherlands
[21]Research Organization of Open Innovation and Collaboration, Ritsumeikan University, Osaka, Japan
[22]Research Fellow of the Japan Society for the Promotion of Science, and [23]Department of Psychology, Education, and Child Studies, Erasmus University Rotterdam, Rotterdam, The Netherlands

(iD) M-MP, 0000-0002-7487-7898; SMF, 0000-0001-7874-1261; PMI, 0000-0002-6922-3590; AEv, 0000-0002-2733-1841; TA, 0000-0002-2387-5219; TD, 0000-0003-3004-8710; RG, 0000-0002-6690-8842; TH, 0000-0003-0565-441X; MJ, 0000-0002-9146-0306; TJL, 0000-0002-5513-6605; RP, 0000-0001-8107-5992; RP, 0000-0002-9085-9876; JMS, 0000-0002-9041-7082; ES, 0000-0003-0836-045X; UST, 0000-0002-6589-3167; MAV, 0000-0003-3413-9062; JMW, 0000-0003-2415-2933; NY, 0000-0002-2656-4432; RAZ, 0000-0001-9967-7879; DvR, 0000-0002-5030-4091

Increased execution of replication studies contributes to the effort to restore credibility of empirical research. However, a second generation of problems arises: the number of potential replication targets is at a serious mismatch with available resources. Given limited resources, replication target selection should be well-justified, systematic and transparently communicated. At present the discussion on what to consider when selecting a replication target is limited to theoretical discussion, self-reported justifications and a few formalized suggestions. In this Registered Report, we proposed a study involving the scientific community to create a list of considerations for consultation when selecting a replication target in psychology. We employed a modified Delphi approach. First, we constructed a preliminary list of considerations. Second, we surveyed psychologists who previously selected a replication target with regards to their considerations. Third, we incorporated the results into the preliminary list of considerations and sent the updated list to a group of individuals knowledgeable about concerns regarding replication target selection. Over the course of several rounds, we established consensus regarding what to consider when selecting a replication target. The resulting checklist can be used for transparently communicating the rationale for selecting studies for replication.

# 1. Introduction

The last two decades have brought uncertainty to the empirical sciences. Researchers have grown increasingly sceptical of the reliability of previously accepted findings, a situation characterized as a crisis of confidence, reproducibility, replication or credibility [1,2]. In psychology, the crisis narrative might have many origins: Ioannidis's controversial article [3], some uncovered scientific fraud cases in The Netherlands [4], the publication of eye-catching findings of extra-sensory perception [5], and a series of methodological papers describing the ease with which results can be covertly pushed into the desired direction (e.g. [5–7]). This narrative has since gained momentum as the centre of fiery debates, has led to a substantial—and growing—body of literature, and has been the catalyst behind the foundation of countless practical initiatives to improve the reliability and quality of empirical, psychological research.

Many in the scientific community have chosen to challenge outdated practices and transform science for the better. While many initiatives aim to dismantle the academic publishing system, or help researchers educate themselves on good scientific practice, other endeavours grapple with problems with the findings themselves. One key element of these efforts involves various forms of replication. Close replications aim to mirror the original study (OS) as closely as possible, allowing for example for better estimates and correction of false positives, whereas conceptual replications change elements of the OS to allow for understanding boundary conditions (e.g. by changing measurement and manipulations) and theory building of a phenomenon. A large increase in articles concerned with theoretical and philosophical discussions on replication and replicability is coupled with a sharp uptick in the number of empirical replication studies being conducted (for numbers until May 2012, see [8]). In psychology, one example is the widespread replication attempt by the Open Science Collaboration, which demonstrated that more than half of the empirical findings under scrutiny did not replicate [9]. Only a third of the original studies (36.1%) suggested a statistically significant effect (i.e. $p < 0.05$) and less than half (41.9%) of the original confidence intervals included the replicated effect size [9].

Increased interest in the discussion and execution of replication studies contributes to the active effort to restore credibility to scientific research, including psychological research. However, it brings with it a second generation of problems. Among these is the fact that the number of potential replication targets is at a serious mismatch with the resources available for replication studies, both in terms of human labour

and in terms of available funds. As one example, in a separate project authors A.E.v.t.V. and P.M.I aim to replicate original research in social neuroscience [10]. Even restricting their candidate set to studies using fMRI in the last 10 years, they currently have a pool of over 2000 potential targets to select from. The rate at which empirical studies in psychology are published has been growing exponentially for the past century. Simultaneously, the rate at which original studies are replicated is very low. The replication rate in social sciences and psychology alike has been estimated at around 1% [8,11], though the rate is difficult to estimate exactly. While the pile of potential replication targets is growing at an exponential rate, funding for replication is developing more slowly. This results in an enormous back-log of non-replicated research to contend with.

To accommodate the need for replication studies, funding opportunities targeting replication studies that have emerged range from broad scale funding opportunities in the Biomedical Sciences (e.g. [12]), Social Sciences and Humanities (e.g. [2,9,13]), or Educational Sciences (e.g. [14]), to specific initiatives calling for replication in pre-specified areas (e.g. [15]). Even so, grants for replications receive many good proposals, but can only fund a low percentage of them. For example, the Dutch funder NWO could only fund around 10% of submitted replication studies [16]. Though there is an increase in the number of funding opportunities, they remain relatively scarce and overall resources for replication studies remain limited.

Another stumbling block in the road toward regaining certainty and credibility through conducting replication studies is the way in which studies are selected as replication targets. As we have argued in recent publications, target selection is haphazard and often poorly motivated (for instance, because replicating authors doubt the veracity of original authors or their findings; see [17]), and does not make the best use of what scarce resources are available [18,19]. Some authors have suggested ways to select replication targets, such as using cost–benefit analysis [20], employing Bayesian decision-making strategies [21] or selecting at random [22]. While at first glance suggestions on how to select replication targets might appear quite different, common themes do exist. In a comprehensive review, Isager and colleagues [16] identified four factors often considered when deciding what is worth replicating: (1) value/impact, (2) uncertainty, (3) quality, and (4) costs and feasibility.

Whatever the reasons for selecting a particular replication target, we believe that communicating how the eventual decision was reached is very important. At present, there is no consensus as to what characterizes a study 'worth replicating' or 'in need of replication'. Regardless of whether or not consensus on this matter can possibly be achieved, clearly communicating one's reasoning behind selecting a replication target enables others to understand, and evaluate the decision. To spend limited resources for replication studies wisely, it is in the interest of both researchers and funding agencies to replicate studies that make sense and that make good use of the resources. Having a transparent logbook of why targets are selected for replication is a first step towards spending limited resources well.

To be clear, we believe that science would benefit from transparently reporting the decisions that led to the genesis of *all* studies. However, we argue that there is good reason to consider the decision process for replication studies separately from original studies. First, the motivation of and reasoning behind replication studies might differ. While many original studies explore new claims based on theoretical reasoning and previous literature, replication studies have in the past frequently been motivated by the intent to corroborate existing empirical results. Second, the room for a replication to add to a field's knowledge base can be more readily quantified since the primary function of a replication is to reduce uncertainty about existing results (whereas original research can have many different functions, some of which are hard to represent quantitatively). Therefore, the selection process may be optimized more easily for replication studies. Third, due to the lack of being able to play the 'novelty card' when justifying the study authors may be facilitated by a systematic approach. With replication and self-correction being deemed important elements of a scientific field [23], a more systematic and transparently documented replication selection process can help characterize—and signal potential points of improvement for—a field's maturation.

To facilitate such a transparent reporting of considerations that led to a replication study, we aim to develop a list of criteria generally regarded as important, which could be used to systematically and transparently justify the selection of a particular replication target. Researchers could use this list to transparently *and* systematically report their replication target selection process, and in turn meta-scientists could use these reports to characterize a field's development. A great example for transparent selection of a replication target was recently published by Murphy and colleagues [24]. While this is a useful start to justifying resource allocation, we believe that we could go a step further by streamlining this process and offering authors structure and guidance in their selection process. Additionally, a list of considerations would offer a structured tool to funding agencies both when

providing money for replication studies specifically and when looking to evaluate the usefulness of a proposal. In the remainder of this paper, we outline how we plan to go about developing this list.

## 1.1. The present study

We argue that the involvement of the wider scientific community is crucial when designing a list of considerations to be used for transparent and systematic replication target selection. In this project, we aimed to (1) describe the considerations generally regarded as important by psychological researchers and (2) construct a list of considerations to be consulted when selecting future replication studies in psychology. To ensure that our results reflect considerations of the selection process generally regarded important by the psychological community, we employed a consensus-based method.[1] More precisely, we used a Delphi approach to expound the considerations and criteria researchers commonly deem important when selecting a replication target.

The Delphi process, which has the goal of developing consensus on a given topic or issue, is one of the most frequently used methods across multiple fields [25]. The Delphi process, as applied in this setting, is descriptive and can be considered an exploratory sequential mixed methods design. It is an iterative process, in which judgements from 'informed individuals' are collected in the form of questionnaire responses. The questionnaire collects both quantitative data in the form of importance ratings and qualitative data in the form of suggestions and opinions on judgements. Over several rounds, consensus on several judgements or opinions emerges [26]. We have chosen this method for use in the current project, as it allows for including researchers from all over the globe, ensures anonymity of responses which allows participants to disagree more freely [25], and is most likely to yield results which reflect the opinions of the group as a whole, rather than capturing the views of a select few outspoken individuals.

We implemented a so-called 'reactive' Delphi method [26]; a modification of the original Delphi method. The reactive Delphi method involves participants responding to a previously constructed version of items, instead of generating a list of items themselves [26]. In the present study, a preliminary list of items was constructed by the organizing authors (M.-M.P, S.M.F, P.M.I, A.E.v.t.V. and D.v.R.) before registration of the project. The organizing authors combined elements from previous suggestions on how to justify replication target selection (e.g. [18,19]) to create a preliminary list of considerations.

A disadvantage of this method is that the quality of the resulting consensus largely depends on the quality of the questionnaire design (i.e. the initial list of considerations; [25]). The authors acknowledge that they might have missed some crucial considerations when constructing the preliminary list of considerations. To overcome this, we included an additional survey round with individuals who selected a replication target in the past. Participants were asked to report how they selected replication targets in the past before judging the preliminary list of considerations. Additionally, participants had the opportunity to suggest additional considerations not yet included. We used the information from the survey to adapt our list of considerations. With this extra step, we hope to have ensured that the questionnaire sent out to the informed individuals contained all relevant elements.

Additionally, the survey enabled insight into the specifics of the selection process and whether it differs depending on the researcher's motivation for conducting a replication and the type of replication. Different considerations might apply to replications that can be more readily termed close replications (e.g. more methodological) than to replications that are more conceptual (e.g. more theoretical). Mapping researcher considerations onto the different types of replications will bring the field one step closer to more explicitly matching outstanding questions for a specific phenomenon with the type of replication that most efficiently answers them (e.g. if an original result is expected to be a false positive, a close replication might be the best match). Although in reality there are many different forms a replication can take (e.g. [27,28]) the distinction between close (direct) and conceptual replication is most common and well known by researchers, which is why for the current survey we examine the relationship with researcher motivations and these articulated ends of the continuum.

Lastly, the updated list of considerations was sent to a selected group of informed individuals, or 'experts', on replication target selection. Over the course of several rounds, participants were asked to judge considerations based on their importance, and given the opportunity to suggest revisions. After

---

[1]We recognize that there may be much disagreement on a local level about what is important—we aim to characterize the opinions of researchers on average, to the extent that is possible.

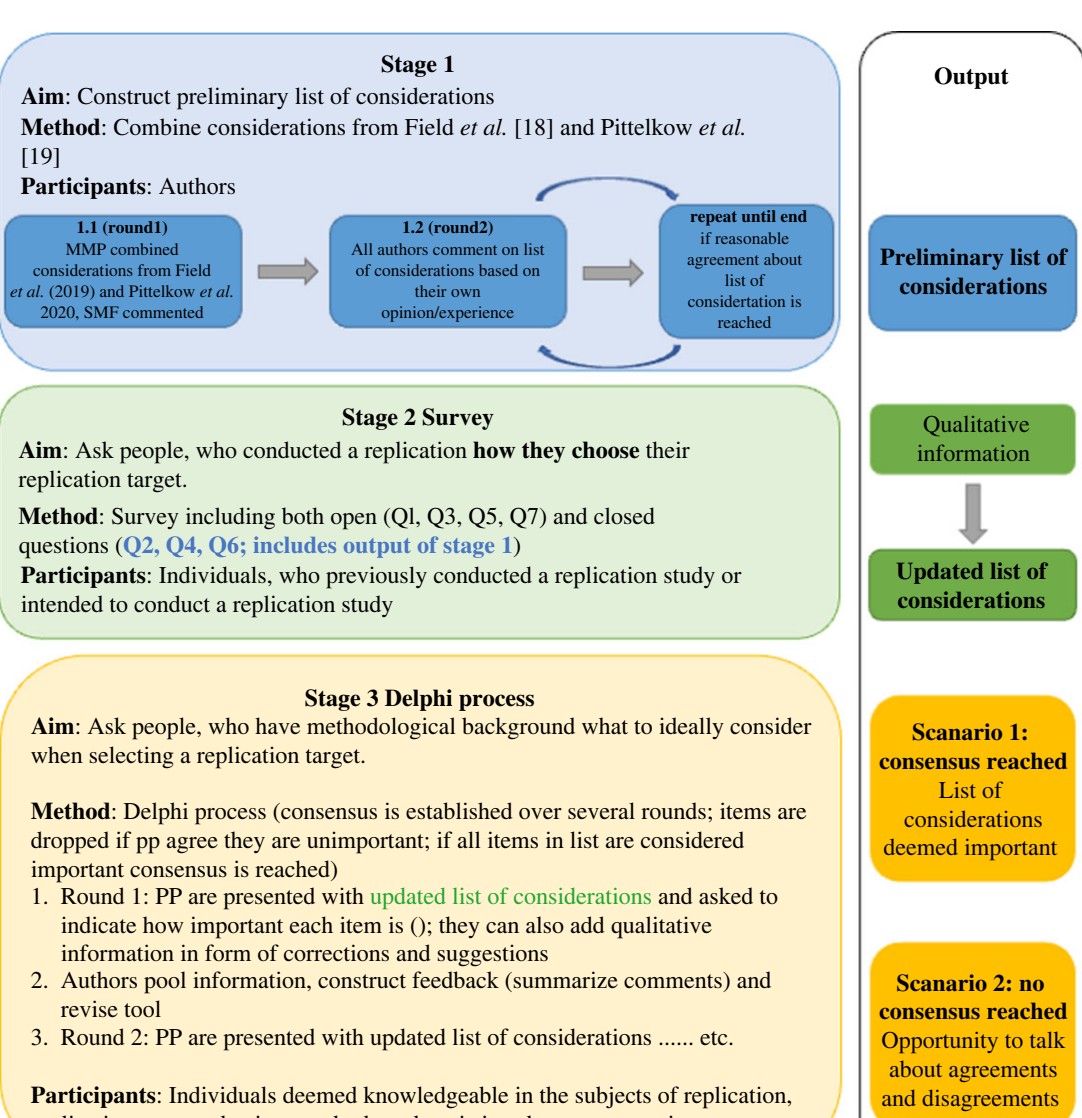

**Figure 1.** Flowchart illustrating the three stages planned for this project.

each round, consensus was evaluated based on pre-specified criteria and participants received a report summarizing the feedback from the previous round. For an overview of the proposed method, see figure 1.

# 2. Methods

## 2.1. Researcher description

M.-M.P. has previously published work on replication target selection in clinical psychology [19]. Her interest in the topic stems from a background in clinical psychology and the realization that sometimes 'shaky' effects are translated into clinical practice. In her opinion, (1) treatments should be recommended only with sufficient evidence, also achieved by replications, and (2) which studies to replicate and how should be determined by evaluating a set of candidate studies. D.v.R. has published theoretical work on replications [18,19,29] and has conducted empirical replications [30,31]. S.M.F. has also published theoretical and empirical works concerning replication [18,30]. Frustration with (sometimes) inefficient use of resources and insufficiently justified reasoning behind conducting replications drives her interest in providing researchers with the means to help systematize the replication target selection process, which can be difficult to navigate. P.M.I has previously authored

theoretical work on replication target selection [16,20]. A.E.v.t.V has previously published on theoretical and practical aspects of conducting replications [32], has conducted large scale and Registered Replication projects [9,33–35], and is involved in theoretical and meta scientific work on replication target selection [16]. Her experience in analysing replications within psychology strengthens her belief that more explicit characterizing of (the process of conducting) replications and their various functions can be a step towards making replication common in research lifecycles and towards theory building through conducting progressive types of replications.

## 2.2. Stage 1

This stage was performed before registration. We created a preliminary list of factors to consider when deciding what to replicate from [18,19]. In a first step, author M.-M.P. extracted themes from these previous publications, and grouped them according to the four themes (1) uncertainty, (2) value/impact, (3) quality, (4) and cost/feasibility identified by Isager *et al.* [16]. Next, author S.M.F. commented on the list and author M.-M.P. adapted it accordingly. Lastly, the organizing authors (M.-M.P, S.M.F, P.M.I, A.E.v.t.V. and D.v.R.) provided feedback and the list was adapted over four rounds until all authors agreed on the final list of 16 considerations. Starting in round three, the authors agreed to not group considerations according to the four themes, as multiple themes applied for some considerations. For example, items grouped under quality, such as sample size, could also inform uncertainty. We will however, after the next stage, ensure that all initial and additional themes will be represented in the pool of items. This first list of considerations can be found in table 1 and the process file is available on OSF.

## 2.3. Stage 2

### 2.3.1. Participants of the survey

We sampled psychological researchers who previously selected a replication target, identified as having either conducted or registered a replication study. We contacted individuals identified through a systematic review of the literature and online search.

We developed a search strategy via pilot searches documented in the electronic supplementary material (i.e. *methods and additional analysis*). Similarly to previous studies [36,37], we identified potential participants by searching the following categories in *Web of Science* using the search string TI = *(replication OR replicated OR replicate)*: Psychology Biological, Psychology, Psychology Multidisciplinary, Psychology Applied, Psychology Clinical, Psychology Social, Psychology Educational, Psychology Experimental, Psychology Developmental, Behavioral Sciences, and Psychology Mathematical.[2] We refined time-span to the last 5 years. To overcome publication bias, we additionally searched the OSF registries using the term *replication OR replicated OR replicate*, again focusing on psychological studies registered in the past 5 years.

Contact information of corresponding authors from eligible articles was extracted. Articles and registrations were eligible if they concerned either a close replication or a conceptual replication in the field of psychology. We defined replications as projects concerning the same effect/hypothesis, independent and dependent variables as specified previous work [27]. In judging eligibility, we mostly relied on the authors' self-presentation. We excluded (1) student projects, as it is unclear whether the replication target was selected or assigned, (2) studies which were clearly not psychology, (3) hits that did not correspond to a research paper or registration, and (4) projects not identified as replications. We were lenient in our exclusion criteria as we expected some self-selection on the side of the participants. This means that we also contacted authors of work where we were unsure whether inclusion criteria were fully met. For eligible registrations, we searched for potential research output and extracted contact information from those records. If no research output was available, we noted (1) the author of the registration, or (2) the author of an associated OSF project (in that order).

The screening procedure is illustrated in figure 2[3] and a full overview is provided on OSF. If the same corresponding author was identified multiple times, we (1) selected the project with clearly met eligibility criteria over one where we were unsure, and (2) selected the most recent project (i.e. the one for which the decision was most recent), as we assumed that participants would be best able to recall the selected

---

[2]Some differences in label terms from [36,37] are due to Web of Science updates.

[3]Please note that figure 2 contains a correction as some duplicates were identified after receiving IPA.

**Table 1.** Preliminary list of considerations constructed in stage 1 and the corresponding item number for the stage 2 survey.

| no. | consideration | corresponding item[a] |
|---|---|---|
| 1 | Do you consider the current strength of evidence in favour for the claim to be weak (as for example quantified by a Bayes factor, a very wide CI, or a p-value close to the typical alpha level of 0.05 combined with a very large sample size)? | Q7 item 12 |
| 2 | Given the current state of investigation of this claim in the literature, how certain are you that the claim is true? Please motivate your answer. | Q5 item 5 |
| 3 | Is the claim theoretically important? If yes, please elaborate. | Q5 item 4 |
| 4 | Do you perceive this claim to have relevant implications, for instance in practice, policy or clinical work? If yes, please elaborate. | Q5 item 3 |
| 5 | Please describe the design of the original study. | Q7 item 2–5 |
| 6 | Enter the sample size. | Q7 item 1 |
| 7 | Who was the sample (for example, what were inclusion and exclusion criteria)? | Q7 item 2 |
| 8 | How was the main outcome measured? | Q7 item 19 |
| 9.1 | Do you consider the outcome measure to be valid? Please motivate your answer. | Q7 item 9 |
| 9.2 | Do you consider the outcome measure to be reliable? Please motivate your answer. | Q7 item 10 |
| 9.3 | Do you consider the outcome measure to be biased? Please motivate your answer. | Q7 item 11 |
| 10 | Do you consider the operationalization appropriate (i.e. are the methods fitted to answer the broader research question that was posed)? | Q7 item 20 |
| 11 | Please describe the analysis plan and performed analysis. | Q7 item 13, 14, 17 |
| 12 | Please enter the observed effect size. | Q7 item 6,7 |
| 13 | Given the sample characteristics, was the sample a good representation of the population? In other words, do the results generalize to the population of interest? | Q7 item 8 |
| 14 | Is the interpretation of the current claim limited by potential confounds? If yes, please describe. | Q7 item 21 |
| 15.1 | Given the original study set-up, is replication readily feasible? | Q9 item 2 |
| 15.2 | Can this study be replicated by generally equipped labs, or are more specific experimental set-ups necessary (e.g. an eye-tracking machine, an fMRI-scanner, a sound-proof booth, etc.)? | Q9 item 1 |
| 16 | How could a replication overcome the issues you raised above? Please also specify the type of replication you intend to run (i.e. close or conceptual). | |

[a]Item numbers refer to the presentation in the electronic supplementary material.

process for most recent projects. In one case, we identified eight projects from one author all published in 2021. In this case, we select one project randomly.

Some of the participants were distant colleagues of the research team. However, the authors did not interact with participants as data were collected anonymously online. Nonetheless, the author names were disclosed during the survey, which might have impacted data collection.

### 2.3.2. Sample size

The survey in stage 2 served as a pilot to inform the list of items provided for the first round of the Delphi process. Sample size determination for qualitative work is complex and depends on a variety of factors such as the scope and nature of the research, the quality of the data collected and what resources are available. Here, we based sample size considerations on the available pool of potential participants. Typically, qualitative studies report between 20 and 30 participants. For the purposes of our project,

**Figure 2.** Flowchart illustrating the identification of potential participants.

we deemed it crucial for our sample to be large enough to be reflective of the consensus in the field. We identified a total of 682 potential participants and with a response rate of 10% we expected our sample to be twice as large as recommended.

### 2.3.3. Procedure

To gain insight into the replication target selection process and pilot the preliminary list of items, we constructed an online survey with 11 questions. The aim of the survey was to (1) pilot the considerations included in the preliminary list and those the author group was undecided about, and (2) to capture considerations not mentioned in the preliminary list. The former was achieved through closed questions rated on Likert scales, and the latter through open questions leaving room for suggestions and additional information. The survey questions are detailed in the electronic supplementary material (*materials and additional data analysis*).

First, we asked researchers to identify the psychological field they work in (closed question). We adapted the sub-field choices from [37,38] and offered participants the choice between: Cognitive and Experimental Psychology, Clinical and Personality Psychology, Developmental and Educational

Psychology, Industrial and Organizational Psychology, Biological and Evolutionary Psychology, Neuropsychology and Physiological Psychology, Social Psychology, Quantitative and Mathematical Psychology, Human Factors, Unsure, and Other.

To gain insight into how replication targets are selected in practice, we asked our participants to illustrate what motivated them to replicate and how they came to pick the particular replication target they chose (open question Q2). Next, participants were asked to describe the type of replication they conducted (open question Q3) and self-identify as either close, conceptual, or other (closed question Q4). To probe our initial list of considerations, we asked participants to indicate to what extent they considered general study characteristics of the OS (closed question Q5), specific study characteristics of the OS (closed question Q7), and feasibility of a potential replication study (closed question Q9). For each of these three aspects, we presented a number of items. On a Likert scale ranging from 1 (*not important at all*) to 9 (*very important*) participants were asked to 'indicate to what extent [they] considered the following pieces of information'. Items represented the initial list of considerations as well as aspects that the authors did not agree upon during stage 1, but which were considered *very* important by at least one author.[4] To avoid ordering effect, items were presented randomly to each participant, such that each participant listed their items in a different order. After each closed question, participants had the opportunity to provide 'any other considerations you had with respect to general study characteristic' (Q6) 'specific study characteristics' (Q8) or 'feasibility' (Q10). Lastly, Q11 provided the opportunity to give general comments and feedback on the survey. To counteract missing data, participants were prompted if they did not answer a question.

Candidate participants identified through the systematic review were contacted via email including a short description of the project and a link to the online survey. The contact email can also be found in the electronic supplementary material (*methods and additional analysis*). We estimated the survey to take approximately 15–20 min. Data collection was open for a month and reminder emails were sent one and three weeks after the initial invite.

### 2.3.4. Data analysis plan

Open-ended items (i.e. Q2, Q3, Q6, Q8, Q10) were analysed using thematic analysis. Thematic analysis is used to identify patterns (themes) within data [39]. During thematic analysis, the researcher plays an active role in identifying, selecting and reporting themes [39]. For the purposes of our project, we used thematic analysis as a realist method, reporting on experiences and judgements from our participants. In contrast to quantitative methods, qualitative data analysis is an inherently flexible and exploratory process, Braun & Clarke [39] mention a number of questions one can consider before data collection, on which we reflected here:

— Themes were identified at the semantic level meaning that we focused on what was explicitly mentioned in the data without examining the underlying ideas and assumptions which shape the content. As such, we consider our analysis to be descriptive.
— We used an inductive, data-driven approach for identifying themes. To this end, we read and re-read the data for themes related to considerations for replication target selection. We were aware that our previous involvement with the topic might impact the themes identified and aimed to be reflexive during the coding process (cf. [40]). Reflections and potential sources of bias were documented. Relevant text from these reflections is discussed in the manuscript, while details can be found on OSF.
— We were interested in extracting the most frequently mentioned themes (i.e. considerations). Prevalence was counted across and not within individuals. In other words, we counted how many individuals mentioned a certain theme, and not how often the theme was mentioned overall. When registering this report, we consciously refrained from quantifying the proportion of participants that need to mention a theme for it to be considered frequent, so that later we were able to judge which themes are the most crucial ones, and in which proportion based on the data. In the result section, we report the number of instances in which themes were mentioned across individuals.
— We were interested in comparing themes between different types of replications. Thus, we contrasted codes identified in responses to Q2, Q6, Q8 and Q10 between different types of replications identified in Q3 and Q4.

[4]Item 16 of the initial list of considerations was the only item not included, because it does not feature a unique consideration for choosing a study to replicate.

Braun & Clarke [39] suggest that thematic analysis consists of six phases: (1) familiarization with the data, (2) initial code generating, (3) theme searching, (4) reviewing themes, (5) defining and naming themes, and (6) producing the report. These phases are not to be performed one after the other; instead the data analysis process is recursive, with the researcher moving back and forth between these phases. Our approach was similar to these broad guidelines. It involves the following preregistered steps:

First, authors M.-M.P. and S.M.F. split the data in two, and worked independently on developing a set of likely codes based on themes identified in the data at this stage. Our approach in this step was consistent with the practice of open coding, that is, we selected chunks of relevant text and associated them with a short phrase or keyword generated from the text itself. Second, authors M.-M.P. and S.M.F. collaborated with one another to determine which codes to include in a codebook. This codebook contained information for each code including a thorough definition of the code itself in the abstract, text snippets as concrete examples, and descriptions of inclusions and exclusions (i.e. concrete cases where a given code might not apply). The codebook is openly available on OSF.

Once the codebook was established, authors M.-M.P. and S.M.F. each went through the qualitative text in its entirety, and coded it according to the codebook. Our unit of analysis was a sentence. Once each person coded the dataset, interrater reliability (IRR) was calculated.

According to Miles & Huberman [41], IRR can be calculated as the total number of agreements (between authors M.-M.P. and S.M.F.) divided by that same numerator, plus the number of disagreements between authors M.-M.P. and S.M.F.. Miles and Huberman suggest that an agreement rate between coders of 80% is sufficient, and we used this same threshold. We had planned to consult a third author (i.e. AvtV), if we had not reached the anticipated IRR. That is, as Syed and Nelson put it, 'one individual's analysis of qualitative data should generally lend itself to be re-captured by another individual who is reasonably familiar with the research question and procedure' (p. 376) [42]. Although replicability is arguably difficult to apply in the context of qualitative research, consistency between coders in this case can certainly be validly applied. IRR performed as a measure of our consistency. Final steps in this process revolved around reviewing, defining and naming themes, as Braun & Clarke suggest.

Closed-ended items were evaluated using the median rating and interquartile range (IQR), a measure of dispersion around the median capturing the middle 50% of observations [43]. Old items with a median rating of 3 and an IQR of 2 or lower were excluded from the list, and new items with a median rating of 7 and an IQR of 2 or lower were included. To explore whether the considerations differed between field of expertise and type of replication, we stratified the sample and compared subgroups.

## 2.4. Stage 3

### 2.4.1. Participants of the consensus process

Panel members were identified using snowball sampling, a type of convenience sampling. Snowball sampling is one of the most frequently employed methods of sampling for qualitative research [44], and especially useful if participants need to meet specific criteria or have certain expertise [45]. First, the research team identifies a number of potential candidates. Next, the identified people are contacted and asked to participate and/or identify others who they see fit to participate in the study. By asking potential participants to consider who else has the expertise needed for the study, snowball sampling taps into social knowledge networks [44], which we considered beneficial to our project as we were interested in shared, communal knowledge regarding replication target selection.

Snowball sampling was implemented as follows: prior to registration, we constructed an initial list of 29 potential participants, who we deemed knowledgeable in the subjects of replication, replication target selection, methods and statistics, theory, or meta-science. The list can be found in the electronic supplementary material (*methods and additional analysis*. To not only identify 'replication experts', but also content researchers, we offered researchers who participated in the stage 2 survey the option to sign up for the Delphi procedure.

Next, we contacted these potential candidates via email, asking them whether they were willing to participate and/or to forward the invitation to someone they might find eligible, and/or to nominate another person by replying to the email. We are aware that this method does not ensure that every potential participant has an equal chance of being selected. To avoid the sample being heavily biased, we attempted to balance participant selection regarding gender, career level, and country of residence.

We planned to make a Twitter call to reach out to members of underrepresented demographic category, relying on 'word of mouth' in the scientific community on Twitter if necessary.[5]

Eligible participants received an online survey, asking them to indicate their agreement with the previously constructed list of considerations on a Likert scale from 1 (*not important at all*) to 9 (*very important*). We also offered the option for free text responses on the phrasing of the considerations and whether important considerations were missing. Quality of consensus is highly dependent on participant motivation. To ensure that our participants were sufficiently motivated, we offered co-authorship in exchange for participation. Authorship was voluntary and not a prerequisite for participation.[6] If Delphi experts decided to identify as authors they were considered *investigators* according to the CRediT taxonomy (see Authors' contributions).

We anticipated the sample to consist (mostly) of researchers who are distant colleagues or perhaps one-time collaborators with some of the author team. Our contact with them in the context of the study was distant.

### 2.4.2. Sample size

Some authors suggest a sample size around 20 members to produce stable results [46,47], while others argue that smaller panels of 6–11 panelists suffice [25]. However, individual responses are very influential in small panels producing potentially unstable results [46]. As the Delphi process is time-intensive, panel attrition is likely. Typically, the overall response rate for Delphi procedures is 80% [48]. Thus, we aimed to recruit a minimum of 30 participants for our study over a maximum period of three months.

We planned that if after one month, our sampling procedure resulted in more than 30 participants, we would proceed to the Delphi process, provided that the sample was balanced with regard to gender, career level and country of residence. Additionally, we planned to stratify participants by their research field similar to [37]. Otherwise, we decided to reject and select participants to create a balanced sample. In the latter case, we planned to report justifications for participant selection. We further planned that if, after three months, our sampling procedure resulted in fewer than 30 participants, we would proceed with the Delphi process but highlight that results might be unstable and recommend replication to establish stability of the considerations. Note that the sample size determination was empirically informed as no clear guidelines for 'optimal' panel size for Delphi procedures exist.

### 2.4.3. Procedure

The goal of a Delphi process is to establish consensus over several, iterative rounds. During each round, participants were asked to judge the importance of a number of items (i.e. considerations) and provide feedback. In between the rounds, participants received structured feedback reports summarizing results from the previous round both quantitatively and qualitatively.

For the first round, participants received the list of considerations, updated by the results of stage 2. For each subsequent round, participants received a revised list of items, for which consensus had not yet been reached. Items were revised according to qualitative feedback from the participants. To define what constitutes consensus and avoid the Delphi process going on indefinitely, stopping rules were implemented. In line with [38] the following pre-specified stopping rules applied: (1) the Delphi process was defined to be 'concluded with unsuccessful recruitment' if three months after contacting potential panel members, there were fewer than six participants; (2) the Delphi process was defined to be 'concluded with consensus' if consensus was reached about the considerations generally regarded as important when selecting a replication target. Consensus was defined as an IQR of 2 or less. Once consensus was achieved for all items, no new round would be initiated; (3) the Delphi process was defined to be 'concluded with incomplete consensus' if consensus was not reached for all items (i.e. $IQR > 2$) after the fourth round. No new round would be initiated after this stopping rule was triggered. We planned to report the last version of list of considerations and highlight disagreements.

---

[5]We acknowledge that such an approach may introduce selection imbalances of its own, however we argue that it is still likely to assist in reaching a wider range of participants.

[6]One participant opted to not be listed as a co-author.

### 2.4.4. Data analysis plan

Data analysis was performed after each Delphi round. Quantitative items were analysed using medians and IQR and the distribution of ratings was visualized using histograms. Items with a median rating of 6 or more and IQR of 2 or less were included in the final list of considerations. Items with a median rating lower than 6 and an IQR of 2 or less were excluded. Qualitative responses were summarized by M.-M.P. and discussed by the author group. We counted how many individuals mentioned a certain concern or suggestion. The list items were revised based on frequently mentioned suggestions. When registering this project, we consciously refrained from defining *frequently a priori* to allow us to flexibly respond to concerns and suggestions later on. We anticipated no incomplete data reports as we forced participants to answer every item. If participants had no suggestions, they were instructed to answer open questions with 'none'. If due to attrition, participants did not join subsequent Delphi rounds, we proceeded with the remaining experts.

After each round of data analysis, M.-M.P. constructed a structured feedback report for the participants. Items for which consensus was reached were not included in the summary report to the participants. In the feedback report we: (1) replied to frequently raised general concerns if there were any, and (2) presented items for which no consensus was reached. For each item, we presented the histogram of responses, highlighted revisions if necessary and addressed item-specific concerns. Summary reports and the invitation for the next round were sent to participants who responded to the previous round.

## 2.5. Reporting of results

During stage 2, we produced: quantitative data (i.e. importance ratings), qualitative data (i.e. participants' responses and corresponding codes), documents containing reflections and potential sources of bias from coding authors, and an updated list of considerations. Quantitative data were summarized using median ratings and IQR and presented in tabular form. We report identified codes and associated frequencies. Reflections of the coding authors and the updated list of considerations are available at OSF. Reflections and reasoning behind what qualified as a theme are discussed in the manuscript, leading to intermediate conclusions about how psychological researchers select replication targets. During stage 3, we produced quantitative (i.e. importance ratings), and qualitative data (elaborations from participants), feedback reports for each round, and a selection of items, which participants agreed upon (i.e. the final checklist), and potentially items that no consensus was reached for. Median ratings and IQR for each item across the rounds are presented in a table. We report our definitive checklist, highlighting in particular the items that reached consensus, but also those that did not. Feedback reports were uploaded to OSF, summarizing also the qualitative input from the Delphi process. These results allowed us to discuss and suggest relevant considerations for future researchers, discuss implications for psychological science, and potentially other social sciences and signal potential direction for future research.

# 3. Results

## 3.1. Protocol and data

All supplementary material including the preregistered manuscript, which received in principle acceptance, informed consent forms, data and analysis files can be found on OSF (https://osf.io/j7ksu/; DOI 10.17605/OSF.IO/J7KSU). Reuse is permitted under the CC-By Attribution 4.0 International license.

## 3.2. Stage 2

### 3.2.1. Deviations from preregistered plan

While we followed the preregistered plan as closely as possible, a few deviations were deemed necessary.

**Stage 2**. First, during data collection 30 additional duplicate emails were identified and removed according to the preregistered protocol. If we had identified two email addresses for one person, we used both to increase the likelihood of a response. Second, despite repeated prompts for participants to answer all items, some data were missing. Some participants indicated why they were unable to

answer specific items, thus providing us with qualitative information about the mechanism of missingness. We therefore considered responses with missing data on some, but not all items, as complete and included it in the quantitative analysis with all data available.[7] Third, we had planned that authors M.-M.P. and S.M.F. would collaborate first with one another, then with the other authors, to determine which codes to include in a codebook. However, the code-book was established by M.-M.P. and S.M.F. without the input of the co-authors. Codes overlapped substantially and disagreements were easily resolved. Lastly, while we meant to exclude all student projects when identifying potential participants, 16 participants indicated that they conducted their replication as student projects. Their responses were included in the analysis as we were committed to use all available data and the respondents were able to describe their decision-making process.

### 3.2.2. Participants

A total of 682 participants were contacted. Of these, 678 individuals were contacted via email on 4 October 2021 using the Google extention GMass. Four additional individuals were contacted by M.-M.P. via LinkedIn on 11 October 2021. Details about the reminders are described in the electronic supplementary material. Data collection was closed four weeks after it had started (i.e. on 1 November 2021).

A total of 185 (27%) responses were recorded. Of these, 64 responses were incomplete, leaving a total of 121 (18%) responses.[8] Demographic information of the 121 responders is presented in table 2.

### 3.2.3. Quantitative analysis

We calculated the median and IQR for all quantitative items. Results are presented in table 3 and visualized in figure 3. None of the items reached our pre-specified decision criterion of a median rating no larger than 3 with an IQR no larger than 2 and none of the new items reached our pre-specified decision criterion of a median rating no smaller than 7 with an IQR no larger than 2. Consequently, we did not change the preliminary list of considerations based on the quantitative analysis.

A second aim of our survey was to examine potential differences in considerations based on the field of expertise and type of replication. To this end, we split the data into different strata and compared the medians and spread of the data (IQR, min and max) for each stratum. The stratified analysis is detailed in the electronic supplementary material (*methods and additional material*). No meaningful differences were observed between sub-fields. Ratings differed slightly between the different types of replication. For example, participants that classified the replication they conducted as *direct* or *close* rated generalizability ($Mdn = 4$), in- and exclusion criteria ($Mdn = 3$) and random assignment ($Mdn = 3$) lower than participants that classified the replication they conducted as conceptual ($Mdn = 7$, $Mdn = 6$ and $Mdn = 6$, respectively). This is most likely explained by the different aims underlying close and conceptual replication. That is, while close replications aim to verify previous findings, conceptual replications aim to generalize findings beyond, for instance, the original study's context or sample. However, participants who conducted a close/direct replication rated statistical error as unimportant ($Mdn = 3$), which is in contrast to the assumptions that the primary aim of close replications is to verify.[9] Nonetheless, differences between subfields and type of replication were not substantial enough to warrant specific versions of the list of considerations for each.

### 3.2.4. Qualitative analysis

First, we split the data in half using 60 randomly generated numbers between 1 and 121. S.M.F. and M.-M.P. independently established codebooks based on 60 and 61 responses, respectively. S.M.F. identified 56 codes, and M.-M.P. identified 67. In two consecutive meetings, S.M.F. and M.-M.P. reviewed and compared their codes and collaboratively established a codebook including 73 codes. Lastly, both S.M.F. and M.-M.P. independently re-coded the complete dataset using the established codebook.

---

[7]Incomplete responses (i.e. when respondents stopped after a number of items) were excluded from the analysis.

[8]The algorithm indicated 66 incomplete responses but two were marked incorrectly.

[9]We received qualitative feedback suggesting that some participants might have misunderstood this question. They meant to indicate that flawed studies should not be replicated (answer: no), where the question aimed to assess whether a study being flawed is a relevant factor for deciding to replicate (which for the above would mean, answer: yes). This limits interpretability of this particular item.

**Table 2.** Number of stage 2 survey participants by their field of interest and type of replication the participant has conducted.

| | total | direct/close replication | conceptual replication | other |
|---|---|---|---|---|
| *N* | 121 | 94 | 17 | 9 |
| psychology field (% per column) | | | | |
| cognitive and experimental | 39 (32.2%) | 30 (31.9%) | 5 (29.4%) | 4 (44.4%) |
| social | 29 (24.0%) | 23 (24.5%) | 4 (23.5%) | 2 (22.2%) |
| clinical and personality | 12 (9.9%) | 9 (9.6%) | 3 (17.6%) | |
| developmental and educational | 7 (5.8%) | 6 (6.4%) | 1 (5.9%) | |
| industrial and organizational | 5 (4.1%) | 4 (4.3%) | | |
| biological and evolutionary | 4 (3.3%) | 3 (3.2%) | 1 (5.9%) | |
| quantitative and mathematical | 4 (3.3%) | 4 (4.3%) | | |
| human factors | 2 (1.7%) | 1 (1.1%) | | 1 (11.1%) |
| neuropsychology and physiological | 1 (0.8%) | 1 (1.1%) | | |
| other[a] | 11 (9.0%) | 10 (10.6%) | | 1 (11.1%) |
| unsure[b] | (5.8%) | 3 (3.2%) | 3 (17.6%) | 1 (11.1%) |

Note: One person did not indicate what type of replication they conducted and was thus excluded from the stratified counts.
[a]Conservation/environmental psychology, differential psychology, experimental analysis of behaviour, human–computer interaction, legal psychology, metascience, parapsychology, psycholinguistic, social and evolutionary psychology, and sociology.
[b]behaviour genetics, communication and media psychology, economic psychology, media psychology, neuroimaging, and sport and exercise psychology.

IRR was calculated as the number of agreements divided by the sum of the number of agreements and disagreements. Agreement was defined as both coders assigning the same code(s) to the same text or assigning the same code to different, but related, text. Disagreement was defined as both coders assigning different code(s) to the same text.[10] The first author noted cases of agreement and disagreement by going through the data case by case and (1) noting clear agreements (same code(s), same text), (2) noting unclear agreements (same code(s), different text), (3) noting clear disagreements (same text, different code(s)), and (4) noting codes only assigned by one coder. A detailed account of this procedure is provided in the electronic supplementary material (*method and additional analysis*). In total, 343 agreements (1), 77 disagreements (2, 3) and 329 quotes identified by only one coder (4) were counted. This resulted in an IRR of 0.82.[11]

The large number of quotes assigned by only one coder might be explained by (1) differences in coding styles (M.-M.P. assigned many more codes than S.M.F. in general), (2) differences in involvement in developing the codebook (M.-M.P. was more involved than S.M.F.), or the coder being more familiar with their own codes as opposed to the one established by the other. The assignment of codes to text involves the interpretation of those texts by the coder; the observed discrepancies are neither surprising nor cause for concerns about validity. To be sure, as Braun & Clarke [49] emphasize, when multiple coders are part of a thematic analysis, the goal is to 'collaboratively gain richer or more nuanced insights, *not* to reach agreement about every code' (p. 55, emphasis in original).

The coders identified two key themes. The first theme, **decision-making process**, describes the process underlying a participant's decision to replicate. In our interpretation, this theme is concerned with *how* participants decided to replicate, and encompasses the aids and obstacles they encountered during the process. The second theme, **motivation**, is concerned with *why* participants chose to replicate a study in general or *why* they choose their specific targets. Themes are not as distinct as we might present them in this text. Motivating factors interact with the decision-making process and vice versa. Below, we describe the themes and their specific sub-themes and relate them to each other. Participants' quotes are presented to illustrate themes and **themes** and important *sub-themes* are presented in bold and italics.

[10]If coders assigned multiple different codes to the same text, this was counted as one disagreement.

[11]If disagreements including multiple codes were counted as multiple, IRR dropped to 0.77.

**Table 3.** Stage 2 survey questions with number of available responses, median rating (1 = not at all important, 9 = very important), and corresponding interquartile range.

| question | N | median | IQR |
|---|---|---|---|
| Please indicate to what extent you considered the following pieces of information when scrutinizing the potential replication target: | | | |
| — Whether the finding has been investigated sufficiently or not. | 119 | 8 | 3 |
| — Whether the citation count of the study was high or low. | 119 | 4 | 5 |
| — Whether the study has relevant implications, for instance in practice, policy, or clinical work, or not. | 120 | 7 | 3 |
| — Whether the finding has a strong connection with theory or not. | 120 | 7 | 3 |
| — Whether the finding was unexpected (e.g. 'counterintuitive', 'surprising'), or in line with what can be expected. | 119 | 6 | 4 |
| Please indicate how important the following specific characteristics of the original study were for you when choosing your replication target: | | | |
| — The total sample size. | 115 | 6 | 4 |
| — Handling of inclusion and exclusion criteria. | 115 | 4 | 5 |
| — Blinding procedures (e.g. blinding of participants, experimenters, analysers). | 117 | 2 | 5 |
| — Sampling procedures (e.g. stratified random sampling, snowball sampling, convenience sampling, etc.). | 115 | 4 | 4.5 |
| — How participants were assigned to conditions (e.g. randomly, single/double blind, etc.). | 116 | 3 | 5 |
| — Statistical power to detect the effect sizes of interest. | 116 | 6 | 4.25 |
| — The size of the effect size. | 119 | 6 | 4 |
| — Generalizability of the sample. | 116 | 5 | 4 |
| — Validity of the outcome measures. | 116 | 6 | 3 |
| — Reliability of the outcome measures. | 114 | 6 | 4 |
| — Potential bias of the outcome measures. | 115 | 5 | 5 |
| — The strength of evidence (measured by reported p-value, confidence interval, Bayes Factor, etc.). | 117 | 7 | 2 |
| — Missing data handling. | 114 | 3 | 4 |
| — Whether the finding was based on within-subject measurements or between-subject measurements. | 116 | 3 | 4 |
| — Open access to underlying empirical data that were analysed. | 117 | 3 | 4 |
| — Whether the study has been preregistered. | 118 | 2 | 4 |
| — Whether the finding was predicted *a priori* or discovered during data exploration. | 114 | 5 | 5 |
| — Whether there are statistical errors in the results reported (e.g. the degrees of freedom do not correspond to the other reported statistics, the total sample size does not equal the sum of the group sample sizes, etc.). | 114 | 4 | 5 |
| — How the main outcome was measured. | 116 | 6 | 4 |
| — Whether the operationalizations were appropriate (i.e. the methods were fit to answer the broader research question that was posed). | 115 | 5 | 4 |
| — Whether interpretation of the results was limited by potential confounds or not. | 114 | 6 | 4 |
| Please indicate how important the following pieces of information were for you when judging the feasibility of your replication study: | | | |
| — Whether the study could be replicated by a laboratory without specialized equipment (e.g. an eye-tracker, a sound-proof lab, an MRI-scanner). | 119 | 7 | 5 |
| — Whether the study concerned a hard-to-collect sample. | 118 | 7 | 5 |

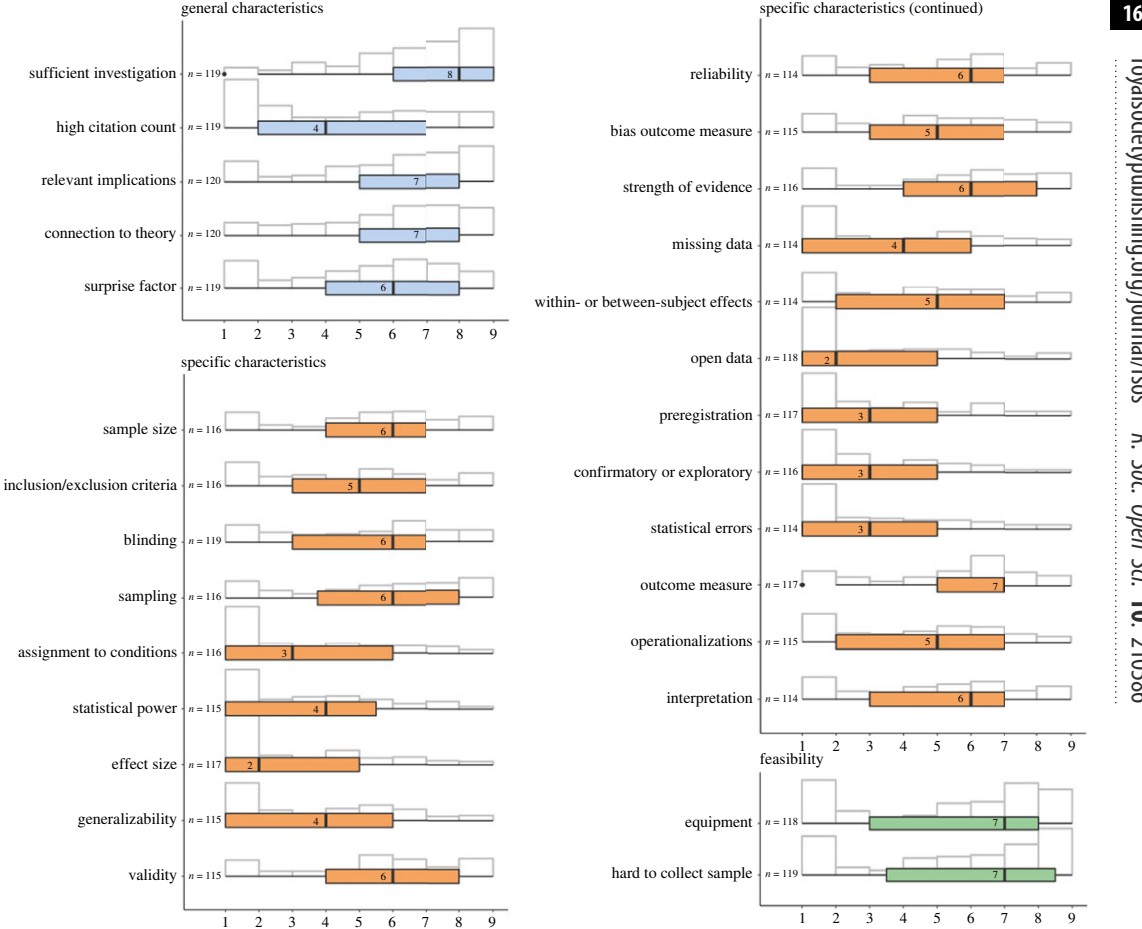

**Figure 3.** Quantitative results. Visual representation of the distributions of responses for all stage 2 survey questions ranging from 1 = `not at all important` to 9 = `very important`. Coloured box plots indicate the median and interquartile range.

*The decision-making process.* To understand *how* our participants decided to replicate an original study, we coded their *process*. We distinguished whether they decided to replicate based on a particular study or whether they decided to replicate before searching for a replication target. However, only 31 participants explicitly described their decision-making process. Moreover, this code was more frequently assigned by M.-M.P. than S.M.F. Interpretation of these results is therefore limited. The mismatch in assignment frequency may reflect M.-M.P.'s specific interest in the process of replication target selection. Participants seemed to more frequently (*n* = 20) decide to replicate *after* reading or conducting a specific study than they decided to replicate before searching for potential replication targets (*n* = 11).

*Institutional influences* shaped the decision-making process for some researchers (*n*=13). Four participants reported being invited to partake in larger replication projects, two of which did not describe their decision-process, presumably as others had made the decisions for them. Other respondents mentioned deciding to replicate for publication purposes. Three explicitly reported changes in journal policies regarding the publication of replications as motivators for them to conduct a replication. Specifically, they replicated original studies previously published in outlets that subsequently incentivized replications. For example, one participant reported that the OS they were interested in replicating was published in a journal that 'had recently adopted policy to publish preregistered replication attempts for their own articles' (case 29) as one factor influencing their choice to conduct a replication.

*Feasibility* played an important role in the decision-making process of our participants (*n* = 76), or as one participant put it 'feasibility was a key issue' (case 115). Feasibility refers to the ease of adapting (if needed), and running the replication study based on clarity and complexity of the OS, as well as the available resources. Feasibility was considered at different points during the decision-making process.

For some, feasibility considerations preceded others, meaning that they only considered original studies which they could run based on their available resources. For example, one participant mentioned that '[they] first considered whether [they] had the skills and resources to run the study' (case 104). For others, feasibility followed other considerations 'After that I selected studies with procedures for which direct replication would be feasible' (case 73). In this way, feasibility was used as a criterion to identify possible replication targets from a pre-selected pool of studies.

To determine the ease of conducting a replication, participants considered whether the method was sufficiently clearly described, and whether implementation of the OS was possible. For some, 'the study needed to have sufficiently detailed description[s] of [the] procedure, instruments and data analysis plan' (case 14). This sometimes coincided with participants mentioning the complexity of the original study's method, or more specifically, the ease with which the OS could be replicated. Participants seemed to look for 'methods [that] were clearly described and easy to implement' (case 82). However, not only studies with sufficient detail were replicated. For example, one participant reported that they 'did not realize how many information about the methods and materials was lacking in the paper' (case 79) until they conducted their direct replication. For some insufficiently provided information was a reason to refrain from direct replication but do 'partial replications because the Method section in the original study was not clear enough on some specifics' (case 94).

Participants further considered the ease with which they could adapt the OS. A few participants specifically mentioned that their replication target was 'easily extendable to additional condition [and], so it was a good fit' (case 81). One specific adaptation considered was whether the OS 'could be translated into other languages or cultural contexts' (case 93). While one might expect this consideration to be more prominent for conceptual replications, it was mentioned in relation to both direct and conceptual replication types.

Related to ease, participants frequently ($n = 18$) mentioned the mode of data collection. Some participants specified the type of data they wanted to collect (e.g. questionnaire or performance data), but participants most frequently mentioned considering whether data was collected on location (e.g. a school or a laboratory) or online, and whether they could adapt the data collection. The need for online data collection was mentioned either as part of the OS methodology 'we only considered studies that were run online' (case 28), or as a possible adaptation 'adapting the method from an in-person context to an online/computerized setting' (case 82). Online data collection might have been a specifically relevant consideration in the context of the COVID-19 pandemic, which prevented many researchers from collecting data on site. For example, one participant specifically mentioned that they 'ensured it [the replication study] could be run online, in covid' (case 119).

Lastly, resources played a large role in considering the feasibility of potential replication targets. Participants considered the degree of overlap between available resources (e.g. time, money, available data, equipment, skills and expertise, and potential collaborators) and the resources required to replicate a specific OS. Participants frequently mentioned time constraints, meaning that '[the replication study] had to be something that I could actually conduct given time and resources' (case 18). Time constraints were often mentioned in relation to financial constraints. Participants either discussed the need to find studies, which could be replicated at 'low costs' (case 75), the need to 'secure enough funding to make it [the replication study] happen' (Case 22), or having 'the funding to support the replication' (case 107). Having access to the data, materials and/or a participant pool, and potential collaborators who would be able to carry out the replication study eased the decision to replicate a specific target study. Lastly, some participants specifically mentioned considering whether they had the skills and expertise to replicate a specific target study. As one participant put it, 'it was important that I had the expertise to perform the replication' (case 107).

Other infrequently mentioned aspects were the ease of getting ethical approval ($n = 2$), participant burden ($n = 3$) and whether the study ought to be multi-sited ($n = 3$).

Naturally, the aspects of feasibility considerations were not mutually exclusive but overlapped within individual participants. For example, available resources would ease adaptation and adjustment of potential replication targets. As one participant described: 'I already had the software for the task, so it was pretty easy to adjust it for the new study' (Case 5).

*Motivation*. Participants' selection of replication targets was motivated by the replicating authors (RAs) **interest** in the original effect, **impact** of the original finding (perceived by the RAs, or objectively demonstrated, e.g. by citations or journal impact factor), **doubt** in the specific effect, specific **methodological aspects** of the OS, or was related to the **author of the OS**. In our interpretation, most participants were motivated by learning from the replication study. For example,

five respondents conducted replication studies to gain familiarity either with the research process (e.g. case 14), or the specific field of research one has not yet encountered (e.g. case 18).

However, replications were not only conducted for personal benefit, but also for altruistic reasons. Ten respondents reported perceiving replications as good scientific practice and thus being committed to running them to 'foster cumulative science' (case 3) or 'establish scientific credibility' (case 55). Others ($n = 16$) conducted replications for educational purposes either as seminar classes, theses, or joined research projects.

*Interest* motivated the majority ($n = 83$) of our participants to conduct a replication study. Many ($n = 32$) specifically mentioned that (aspects of) the OS interested them and motivated their decision to replicate. Participants called it 'interest in the topic' (case 3) or simply stated 'the study we chose was interesting' (case 58), sometimes also labelling it as 'curiosity' (case 10). Three participants said they were interested in participating in the scientific discussion rather than aspects of the OS *per se*, and used involvement with a replication study to do so.

Participants mentioned several areas of interest, the most frequent ($n = 34$) being the motivation to verify the literature body. Many ($n = 13$) participants were planning to conduct their own experiments in the line of the OS, but wanted to verify the validity or reliability of the effect they aimed to extend first. Three other respondents were specifically interested in verifying the paradigm used in the OS as they were planning to use it in their own research. However, verification of the literature body was not always self-serving. Some ($n = 5$) specifically mentioned the motivation to verify the literature body to foster knowledge or explore robustness of the effect. Respondents mentioning the motivation to verify the literature most frequently ($n = 30$) conducted close replications. This is in line with our assumption that the function of close or direct replications is to verify existing research.

Related to, and overlapping with, the motivation to verify the literature body, many participants ($n = 19$) reported an interest in self-replication. This meant that participants repeated their own studies either because it was standard practice to them—'Typically, we (our lab) provide replication studies *within* the original papers' (case 11)—or to verify their own findings. Verification could be motivated by methodological shortcomings. For example, one respondent noticed that 'the results was on shaky ground for some methodological shortcomings' (case 62). Most frequently, however, our respondents wanted to ensure that their findings were robust, valid and stable.

If not their own studies, respondents were frequently ($n = 17$) interested in replicating OSs that were relevant to their own line of research or that they were familiar with. One participant explained that replicating studies familiar to the researcher was attractive because it was relatively easy: 'had conducted a previous study with similar methodology and knew that [they] could easily do another, similar study' (case 118). However, mostly respondents opted to replicate studies that were 'influential to [their] ongoing research program' (case 82). Within one's line of research, interest was also sparked by novel methods, tools or measures. Sometimes, novelty coincided with 'striking' (case 119) findings. Other times, the OS 'broke very new ground' (case 96). As one participant put it 'we felt that something that novel and unexpected […] should be replicated' (case 96).

Likewise, the context of the OS interested some respondents ($n = 14$). Participants were interested in context-dependency of the original effect or how changes in cultural and societal context might have impacted the original findings. For example, one participant 'was finding different results in another context and wanted to understand the phenomenon better' (case 34) and thus explored the context-dependency of the OS. Another respondent postulated that 'the results might be different in a sport context' (case 39). Similarly, some ($n = 5$) respondents specifically mentioned interest in exploring the boundary conditions of the original effect.

**Impact** of the OS was mentioned by 61 respondents. Our participants replicated studies they judged to be generally important or 'seminal' ($n = 24$, e.g. case 27, 87), to the field. For example, one participant explained that the replication target 'was a study that had had a considerable impact on our field' (case 61). Impact was sometimes defined as 'a lot of people talking about it' (case 81) or 'a lot of labs doing conceptual replications' (case 61), or a study pioneering a method not commonly used in the field of research. Overall, it appeared that our respondents were motivated to replicate cornerstone research, which was perceived as most valuable if the replication had impact regardless of the outcome.

Additional qualifiers of impact were citation count ($n = 20$) and the journal that the OS was published in ($n = 10$). As one participant put it, 'we choose to replicate [the OS] because: […] it is an influential finding, as the original article is a well cited paper, published in a high impact journal' (case 85). Another respondent identified the OS as part of the scientific discourse, and therefore important to replicate, as it 'was published in a high ranking journal and […] cited multiple times' (case 21). It appears that citation count and impact factor were used by many participants to judge the impact of an OS.

Studies were also identified as impactful by participants if the conclusions had theoretical relevance ($n = 19$). Replication was believed to 'provide insight into the credibility of […] theory' (case 93) or enable participants to 'weigh in on a larger theoretical debate' (case 111). There was some discrepancy as to the role that theory played in the decision-making process. While theory could be regarded 'as unimportant, because presumably the theory that underlies replication targets is weak to begin with' (case 10), theory was also specifically mentioned to be 'powerful and […] well specified/falsifiable' (case 7). It appears that there is no consensus as to whether studies with weak or strong theory ought to be replicated.

Eleven participants also considered the impact of the replication study instead of the OS. Respondents were motivated to replicate studies 'for which in the past no direct evidence was available' (case 4) or which were judged by them to be 'understudied topic[s]' (case 49). Respondents appeared to assume that replications could serve an important role if the evidence regarding the original finding was limited. However, one participant cautioned that 'a study may not be worth replicating simply because the phenomenon under investigation is understudied—there may be a reason why few studies have been conducted on a particular topic (e.g. little to no clinical or theoretical merit)' (case 106).

Impact outside of the academic discourse was also considered by nine respondents. Specifically, the impact of the original finding on society or policy and the public interest in the original finding. Though one mentioned that they did not care about policy implications (case 109), the other eight were motivated by the practical importance of their replication study.

**Doubt** motivated 62 of our respondents to replicate a study. Doubt means that the RAs believed that they had reason(s) to be sceptical regarding the 'truthfulness' of the original finding. This was mostly ($n = 22$) due to potential flaws of the OS. Some respondents suspected the original finding to be 'due to design error or confound' (case 5) or 'the original study [to have] a series of methodological and statistical flaws that called the results of the original study into question' (case 13). As one respondent put it 'the [original] result was on shaky ground for some methodological shortcomings' (case 62), thus motivating replication to overcome said shortcomings. Interestingly, potential flaws were mentioned for both close and conceptual replications, though it stands to reason that in either case participants modified the original methodology to overcome shortcomings.

Seventeen sources expressed doubt in the original finding based on how 'surprising' they perceived it to be. While novel findings can be surprising (see, for example, the account in case 25), this code is distinct in that respondents clearly mentioned their disbelief in the original findings, which was not necessarily true for novel findings *per se*. Respondents were surprised by findings 'that were different from what one would expect from general experience' (case 8), that is, they were 'unexpected/ counterintuitive' (case 32). Replicating the surprising findings was a way to 'ensure that the conclusion was right' (case 37). It appeared that some participants were more inclined to replicate studies for which they did not believe in the finding. One respondent made this explicit, saying: 'in general, I look for papers that I don't believe the findings' (case 78). This is in contrast to those who are interested in replicating to build on the original finding.

Doubt could also be due to the statistical evidence appearing weak to the participant ($n = 15$). This could be due to small sample sizes, weak methodology, large effect size and associated confidence interval, high p-values, weak statistical evidence as measured through Bayes factors, or peculiar statistical analysis. In some cases, concerns about the statistical evidence coincided with concerns about potential questionable research practices (QRPs). Respondents mentioned p-values showing 'peculiar pattern, with many p-values close to the significance threshold' (case 35) or that 'the initial statistics were very p-value based (indicating a desire to get a $p < 0.05$)' (case 98). Others mentioned 'analytical creativity' (case 104) causing doubt. Additionally, respondents mentioned no analytical reproducibility, preregistration or sample size planning, all of which called into question the original finding and motivated (mostly close) replication for the participant.

Failed previous replication attempts further motivated 14 participants to replicate. Respondents mentioned trying to build on the OS, which included an initial replication of the original effect that failed. Consequently, they decided to run a planned replication instead. For example, one participant 'tried to follow up the work [the original authors] did and so first replicated it. Because the replication failed (non-significant results), [the RAs] tried again' (case 21). Another respondent shared that they 'tried to build on a new and interesting finding but after several attempts found no effect at all. That is when one of [their] co-authors suggested to go back to the original study and try to replicate that first' (case 45).

The lack of replication studies or replications outside the original author's lab similarly caused uncertainty and doubt about the original effect in some participants' minds. The lack of 'internal or

external' replications resulted in the original finding not appearing convincing (e.g. case 9). Still, only internal replication (i.e. as opposed to external corroboration) could also raise reasonable doubt (e.g. case 7). Respondents also argued that the lack of previous replication studies made it 'easier for reviewer to see the relevance of a replication' (case 13).

Respondents ($n = 13$) also mentioned doubt if the original finding was not in line with the current theory or if the literature provided mixed support for the effect. This was true for older studies, which were not further supported by more recent data or novel studies calling into question the current theory. Respondents mentioned the finding being 'out of line with existing work' (case 41) as a motivation to replicate. It seemed that the participants were interested in verifying the original finding before trying to explain why the effect was not in line with the literature or theory.

Lastly, issues with the original author made some respondents doubtful about the original finding. Respondents expressed doubt if 'the author was ambiguous when [they] asked them for help' (case 15) or were not willing to share their data or materials. A few respondents ($n = 3$) also explicitly cared about the original author's reputation, though another respondent stated that they 'do not care about […] author' (case 109). However, for one participant, the reputation of the original author even increased confidence in the original effect: 'we knew the original author and found him trustworthy' (Case 76). Similarly, many participants ($n = 20$) mentioned cooperating with the original authors, which for some was explicitly positive. For example, one participant mentioned that they 'were able to run [their] replication effort thanks to the willingness of the original author to share their data, stimuli, and instructions' (case 77).

*Methodology*. Participants ($n = 77$) mentioned several methodological aspects of the OS motivating their decision to replicate, with some ($n = 8$) making their decision to replicate contingent on specifics of the original method (e.g. 'needed to be carried out with child or adolescent participants', case 14). Sample size was the most frequently ($n = 26$) mentioned concern. Respondents mentioned the original sample being 'rather small' (case 9), criticized that the original sample size had not been justified, or expressed their motivation to collect a larger sample. Sample size concerns could be linked to concerns about the effect size of the OS. Respondents specifically mentioned studies with small sample and large effects being in need of replication. Moreover, these concerns were amplified if the study was not preregistered. For example one participant judged that their target finding 'did not seem very credible (small N/large effects sizes/not preregistered)' (case 114).

Respondents ($n = 16$) were also concerned with the generalizability of the OS. Generalizability means that RAs examined whether the original finding would extend to different stimuli, settings, or populations. Consequently, generalizability was a frequent concern for replicators, who already had access to a different population than the OS. This code further connected to participants mentioning the demographics of the target population for their replication. For example, one respondent said that their 'replication used very similar methodology, but extended the research question to a different population with greater representation of the clinical symptoms [they were] interested in studying' (case 49). It appeared that some respondents found replications especially valuable if they could examine a population different from the OS. One participant made this explicit saying that '[they] also had the opportunity to collect data from a population demographically different from the original study, increasing the value of the replication' (case 72). However, another participant judged it important to use 'a sample as similar as possible' (case 85). Notably, most respondents concerned with generalizability and extending the effect self-identified as conducting close replications.

Methodological aspects of the OS could induce doubt in the 'truthfulness' of the original finding. Outdated methods were frequently mentioned ($n = 9$). In some instances, outdated methods prompted doubt. For example, 'advances i[n] methodological sophistication and quality prompted reconsideration of prior findings that were published using, now, outdated methods' (case 30). Other times, outdated methods did not induce doubt but were considered when updating the methodology to fit the current context. For example, 'the statistical analysis we used was updated to reflect advancements in the capabilities of statistical software' (case 42) or 'we used updated and better validated measures' (case 32).

Respondents were further concerned with potential confounds biasing the original finding. Participants 'chose […] [the] study because [they] thought there was a confound in the experimental design' (case 38) and consequently controlled for 'a factor the original authors hadn't' (case 94). One explicitly mentioned confound was experimenter bias. For example, respondents worried about the potential influence from experimenter bias which leads to 'doubt about methodology' (case 103) or as another respondent put it: '[…] I was afraid that the original study was suffering from experimenter bias' (case 4). Similarly, this prompted participants to replicate with updated methods.

Respondents ($n = 5$) further mentioned statistical significance as an 'implicit criterion' (case 14). Participants were mostly interested in replicating studies for which 'results supported the hypothesis' (case 93), though one person explicitly mentioned 'the null result' (case 19) as motivating their choice to replicate.

Methodological aspects could also be linked to feasibility considerations. More specifically, some participants ($n = 9$) mentioned that they were interested in replicating simple studies specifically, 'which could be replicated easily and quickly' (case 18). This criterion was predominantly applied to student projects.

Infrequently mentioned considerations included the number of trials ($n = 3$), practising specific statistical analyses ($n = 2$), or replicating the OS with the same sample as previously used ($n = 1$).

### 3.2.5. Limitations

Results from the survey need to be considered in light of some limitations. First, some participants misunderstood the instructions and answered the items with replications in general in mind instead of the specific replication study that was the basis for us approaching them. This means that some participants reported concerns that were more general and broad. This might account for some discrepancies and some of the variability in the ratings. For example, participants might simultaneously (1) believe that replication should be concerned with generalizability *in principle*; (2) have not considered it a relevant aspect in the decision to conduct their own replication study.

Asking participants to classify their own study as a direct/close or conceptual replication also means that many people will have applied labels according to different criteria or based on different understandings of the concepts of direct/close and conceptual replication. For instance, many participants that conducted their replication study (partly) to extend the original design or to include additional conditions classified their study as a close replication (with extensions). Nonetheless, one could argue that these cases could be classified as conceptual replications. Our results highlight the variability in replication aims and procedures, and the fact that names and definitions are used somewhat interchangeably and vaguely in the literature. In our view, the dichotomous distinction between the two types of replication is not very informative. Defining replication types based on what they might achieve, or going even deeper [50–52] might be a better approach.

### 3.2.6. Changes based on survey

The most frequently reported codes were identified by counting how often themes were mentioned across cases (i.e. how many participants mentioned a code). Counts ranged from 1 to 34 with $Mdn = 10$. Codes with 10 or more mentions ($n = 38$) were evaluated by the author team. Authors M.-M.P., P.M.I, A.E.v.t.V. and D.v.R. read through the list of frequently mentioned codes, tried to identify connections, linked them back to the preliminary list of considerations and suggested edits.

M.-M.P. and D.v.R. independently summarized the suggestions and both created a suggestion for a revised version of the list of considerations each. M.-M.P. merged the two suggestions and created a first draft of the revised list. Over the course of three rounds, this draft was further revised by the author team with M.-M.P. summarizing co-authors' feedback between rounds. The intermediate list revisions are detailed in the additional files provided on OSF (*List revisions*).

The revised list included 18 items clustered around the six most frequently mentioned themes: (1) interest, (2) doubt, (3) impact, (4) methodology, (5) feasibility, and (6) educational value. These themes partially overlapped with the four themes considered during stage 1, namely uncertainty (here doubt), value/impact, quality (here methodology), and cost/feasibility. Table 4 contains the 18 items (i.e. the rows that have an entry in column 'Round 1').

# 4. Stage 3

## 4.1. Deviations from preregistered plan

After five weeks of data collection, data of 32 respondents was downloaded. However, five responses were empty, leaving a total of 27 participants. We had initially planned to continue recruitment for three months or until reaching 30 participants. However, in light of the fact that the summer months were coming up, we decided that it was better for the quality of the data to proceed with the Delphi process rather than wait two more months for the last three participants to potentially join. We requested permission for this deviation from the editorial office and received approval on 7 June 2022.

**Table 4.** Quantitative results of stage 3 checklist development over the course of the three rounds. Included items are highlighted in bold and revisions are indicated in italic.

| item | round 1 | | round 2 | | round 3 | | decision |
|---|---|---|---|---|---|---|---|
| | Mdn | IQR | Mdn | IQR | Mdn | IQR | |
| **the relevance of the original study for your current line of research or the field you work in** | 7 | 2 | | | | | include |
| your involvement in the line of research that the replication target is concerned with (e.g. self-replication, planning to build on the study in the future) | 6 | 3 | | | | | revise |
| *the degree of involvement you have in previous or upcoming projects related to the replication target (e.g. self-replication, planning to build on the study in the future)* | | | 6 | 3.5 | | | revise |
| *your personal stakes in the replication target's results (e.g. self-replication, financial stakes or other potential conflicts of interest, planning to build on the replication target results in future research, etc.)* | | | | | 7.5 | 3 | no consensus |
| **the current strength of evidence in favour of the original claim (e.g. a high/low Bayes factor, a wide/narrow confidence interval, a high/low p-value)** | 7 | 1 | | | | | include |
| your personal belief about the truthfulness of the original claim (e.g. consensus in findings, replication attempts) | 5 | 2 | | | | | exclude |
| your expectations about whether the original claim would replicate or not | 5 | 2 | | | | | exclude |
| **the importance of the original study for research (e.g. often/rarely cited, under/over-studied, published in high/low impact journal)** | 7 | 1.5 | | | | | include |
| **the theoretical relevance of the original claim** | 8 | 2 | | | | | include |
| **implications of the original claim (e.g. for practice, policy or clinical work)** | 8 | 2 | | | | | include |
| the clarity and replicability of the original protocol (e.g. completeness and clarity of the methodological description, accessibility of the materials) | 6 | 4 | | | | | re-evaluate |
| — | | | 4 | 4.25 | | | revise |
| ***the (un)clarity and (un)replicability of the original protocol (e.g. completeness and clarity of the methodological description, accessibility of the materials)*** | | | | | 7.5 | 2 | include |
| **the sample size of the original study (too small or too large)** | 7 | 2 | | | | | include |
| **flaws of the original design (e.g. in- an exclusion criteria, potential confounds)** | 8 | 1.5 | | | | | include |
| operationalization of the original study's measures (e.g. validity, reliability, and bias) | 7 | 3 | | | | | re-evaluate |
| — | | | 7 | 2.25 | | | revise |

(*Continued*.)

**Table 4.** (Continued.)

| item | round 1 | | round 2 | | round 3 | | decision |
|---|---|---|---|---|---|---|---|
| | Mdn | IQR | Mdn | IQR | Mdn | IQR | |
| *operationalization of the original study's measures (e.g. validity, reliability, and bias) and how this impacts the credibility of the original study* | | | | | 7 | 1.25 | include |
| **concerns that questionable research practices have been employed (e.g. presence/absence of preregistration, potential of p-hacking or HARKing)** | 7 | 2 | | | | | include |
| **generalizability of the original finding (e.g. cultural and temporal context, representativeness of the sample)** | 7 | 2 | | | | | include |
| **the resources available to you for replicating the original study (e.g. funding, time, equipment, study materials, or data)** | 8 | 2 | | | | | include |
| the adaptability of the original study design (e.g. mode of data collection, whether the study can be translated into other languages, contexts) | 6 | 2.5 | | | | | re-evaluate |
| — | | | 6 | 2.25 | | | revise |
| *the adaptability of the original study design (e.g. whether data is collected online or on-site, whether the study can be translated into other languages or applied to different contexts, etc.)* | | | | | 6.5 | 1 | exclude |
| your previous experience and expertise with regards to the original study | 5 | 4 | | | | | revise |
| *you (i.e. all replicating authors) previous experience and expertise with regards to the original study* | | | 5.5 | 3 | | | revise |
| **your (i.e. the replicating team as a whole) presence or absence of previous experience or expertise on the original study as a practical concern** | | | | | 7 | 2 | include |
| educational value of conducting the replication study (e.g. for a thesis or student project) | 5 | 3.5 | | | | | re-evaluate |
| — | | | 3 | 4 | | | re-evaluate |
| — | | | | | 5 | 1.5 | exclude |

Note: Re-evaluate means that participants received qualitative feedback and were asked to rate the same item again.

The consensus procedure was stopped after three instead of four rounds, even though we did not reach consensus on one item. Specifically, we observed diverse responses with very little movement between rounds despite revisions of the item (Round 1: $Mdn = 6$, IQR 3, Round 2: $Mdn = 6$, $IQR = 3.5$, Round 3: $Mdn = 7.5$, $IQR = 3$). We reasoned that burdening participants with an additional survey round would not lead to consensus on this item. We requested permission for this deviation from the editorial office and received approval on 13 September 2022.

## 4.2. Participants

A total of 63 participants were contacted and invited to participate in the Delphi procedure on 25 April 2022. In addition to the 29 potential participants *a priori* identified, 34 survey participants indicated interest in participating. We received 27 responses in the first round, and 20 in the second and third round. During the third round, four participants responded twice. We followed up with

these participants and included the response, which they identified as most closely reflecting their opinion.[12]

Participants were diverse across career stage, field of expertise, gender and geographical location. Participants included five PhD candidates,[13] three post-doctoral researchers, 11 senior researchers and one independent researcher. Participants stemmed from various (psychological) fields including psychological methods and statistics, cognitive and experimental psychology, social psychology, clinical and personality psychology, legal psychology, but also philosophy, empirical aesthetics and (cognitive) neuroscience. Participants identified as men, women, or other. Geographical locations were diverse, but we were unable to recruit participants from South America, Africa, Australia, or the Caribbean or Pacific Islands.

## 4.3. Results

Overall, three Delphi rounds were conducted. Table 4 summarizes the quantitative results and qualitative changes across the three rounds. Detailed summary reports sent to the participants between rounds can be found on OSF.

During the first round, consensus was established for 12 out of 18 considerations. Based on the preregistered criteria 10 considerations with a median rating of 7 or higher and an IQR of 2 or lower were included, and two considerations with a median lower than 7 and an IQR of 2 or lower were excluded from the final list. No consensus was reached for the remaining six items. Two out of the six items were revised based on the qualitative results.

During the second round, we did not reach consensus for the remaining six items. However, the qualitative input allowed us to revise all items as well as provide some clarifications regarding the aim of the checklist. Specifically, we clarified that the aim of the checklist is to transparently communicate one's rationale for selecting a particular study and not whether a study generally needs to be replicated or not.

During the third round, consensus was established for all but one item. Based on the preregistered criteria three considerations were included, and two considerations were excluded from the final list. No consensus was reached for the remaining item and responses were particularly varied ranging from 1 = not at all important to 9 = very important. As a result, this item is not included in the final checklist.

The final checklist included 13 out of 18 items centred around the topics: interest, doubt, impact, methodology and feasibility. See the appendix for the final checklist or download the checklist from https://osf.io/jd9th.

# 5. Discussion

## 5.1. Checklist for transparent reporting of replication target selection

Our goal was to develop a checklist for transparent and systematic reporting of the process of replication target selection. Our consensus-based checklist was designed to guide social scientists through the process of selecting a replication target study, and give them a framework for reporting their decisions and justifications. Checklist item selection was informed by two sources: (1) scientists' practices, revealed by a qualitative analysis of survey data, and (2) expert opinions, explored through a Delphi panel discussion.

Importantly, this checklist covers reasons why a study *was actually selected*, not a list of reasons why a study *ought to be selected*. That is, rather than reporting whether a study needs to be replicated in general, the checklist aims to transparently communicate one's rationale for selecting a particular study. We initially planned to create a list of items which ought to be *ideally* considered when selecting a replication target (see also the specification in figure 1). However, the survey illustrated the variety of potential reasons to select a replication target and underscored the need for transparency, more so than validity of the items. For example, while some might consider it invalid to replicate a study because it was easy to do (the relevant know-how was already present in the team), this reasoning frequently informed replication target selection in practice. Consequently, we moved away from what

---

[12]An analysis with all responses is presented in the stage 3 summary report on OSF.

[13]One PhD candidate is also a Research fellow.

to ideally *consider* towards what to ideally *report*. Specifically, we asked our Delphi participants to consider that if the researcher used a consideration as a ground for replicating (irrespective of their personal assessment of the legitimacy of that reason), was it important for that reason to be explicitly communicated? The checklist can either be used to compare several targets for replication in an attempt to identify and justify the chosen replication target, or to report the justification for having chosen a specific replication target after the fact.

We argue that our checklist will enable evaluation of future decisions to replicate and aid discussion about how resources are allocated, and which studies ought to be prioritized. Our checklist will also be useful to assist replicating researchers in explicating their decision process as they prepare their study protocol. The checklist could also be used to evaluate funding applications for replication studies. For the purpose of justification and decision-making, we advise researchers to complete this list before the start of a replication project. For the purpose of documentation, researchers might complete this list at a later time point. However, we caution that hindsight bias might affect the accuracy of the information if the checklist is filled out after the project is complete.

## 5.2. The guiding principles of replication target selection

Checklist items are grouped according to five themes that we constructed from the survey data: (1) interest, (2) doubt, (3) impact, (4) methodology, and (5) feasibility. This theme structure is validated by similar findings in the literature, such as those of Isager and colleagues [16,53]. Reviewing 68 self-reported justifications for replication target selection, Isager [53] identified four factors guiding replication target selection: (1) uncertainty, (2) value/impact, (3) quality, and (4) feasibility. While we initially adopted the structure proposed by Isager and colleagues [16], it was abandoned during stage 1 as we were unable to clearly group items to one theme or another. Seeing that we independently reconstructed these themes during the present survey lends further credibility to them being the guiding principles of replication target selection. Note however, that we cannot exclude the possibility that we surveyed some authors whose replications were also reviewed by Isager [53]. A quick search demonstrated that some of the potential survey participants we identified were also listed in the Curated Replications Table on curatescience.org. Nonetheless, the present survey included more potential participants of replications published after 2017[14] than before, so potential overlap should be minimal. Moreover, in the present project the qualitative analysis was performed by M.-M.P. and S.M.F., without the input from P.M.I.

We identified four stable principles that likely underpin replication target selection: *doubt/uncertainty*, *impact/value*, *methodology/quality* and *feasibility/cost*. These are complex constructs, whose meaning and interpretation include several factors, as illustrated by the nested structure of the checklist for transparent replication target selection. Still, researchers looking to strategically choose which study to replicate can use these themes to guide their decision-making process. For any study considered for replication, researchers might ask: (1) Is there reason to doubt the findings? (2) Is the topic important? (3) Are the methods capable of saying something meaningful about the topic under study? (4) Is it feasible to replicate the study in a way that will meaningfully reduce doubt about the findings? We argue that since all four factors interact in generating replication value (for a formal definition of replication value, see [16]) the answer to all four questions above should be 'yes' before a replication is undertaken.

While these principles are a good starting point, each researcher still needs to decide what it is that makes a claim doubtful, have impact, speak to the underlying research question(s), and its methods feasible to be attempted again. The checklist we constructed yields a transparent strategy to select a replication target and guides researchers through these four principles, while providing pointers on how to assess them to avoid arbitrary decisions. This might counteract one notable if unwelcome feature of the replication movement in the 2010s—the contentious atmosphere in channels such as society publications and social media [54]. Specifically, some proponents of replication have taken a maliciously gleeful tone in greeting non-replications, while replication efforts have conversely been disparaged as motivated by hostility and destruction. While explicitly hostile motives were unlikely to emerge from our method based on self-generated explanations of replication research, the controversy does point to the need to clarify the prescriptive grounds for the decision to replicate. For example, doubts based only on hunches or suspicions may cover up inadmissible biases, and it is better to base doubt-based selection on clearly expressed arguments from prior theory or evidence.

---

[14]the cut-off time for [53].

The checklist, when used for transparent reporting, can further shed light on the weight placed on each factor by the RAs. It does not prescribe how to judge each of the items allowing for subjectivity and variability between researchers and contexts to enter the process of replication target selection. This might help to develop individualized strategies for deciding what to replicate, each serving different interpretations of what 'uncertainty', 'value', 'quality', 'cost'—and hence, 'replication value'—means. This in turn could inform the definition and quantification of replication values.

We identified two additional guiding principles, which were comparably less stable: *interest*, and *educational value*. Personal interest, also mentioned by Isager [53], was frequently mentioned as an internal motivating factor during the survey. However, expert opinions differed as to whether this item *should* play a role in the decision-making process. Respondents agreed that the relevance of the original study for the RA's line of research plays a role in replication target selection and ought to be communicated. However, they were conflicted about the nature and importance of the RAs' involvement in the original study. Some argued that the RAs should not 'need to have a personal investment in the outcome/line of research' or considered personal investment as harmful as 'it is also important that the research is designed and conducted impartially'. Others argued that 'scientific and societal stakes should supersede any personal stakes'. However, it appears that personal interest plays a role in replication target selection in practice. Indeed, we cannot assume that scientific stakes will be at odds with personal stakes in cases where personal interest (partly) motivates a replication target selection, especially given that many people's personal interests involved the belief that the OS was interesting, important and worth reinforcing with replication. Moreover, replication context aside, personal interest is a common reason for a researcher to select any given research topic [55], and, some of us argue, a valid one. Should we constrain replication target selection such that personal interest is not part of the decision-making process? We argue that providing a transparent report of the decision-making process in replication target selection largely mitigates the potential risks of allowing personal interest as a motivation for replication.

Some participants reported that in their experience 'self-replication was indeed a strong and primary motivation' or suggested that 'it is important that researchers are also invested in replicating their own work'. As a result, it 'would be important to disclose conflict of interest […] as it might point to bias'. Ultimately, no consensus was reached for this particular item. Controversy may nonetheless be a good reason for RAs to report their personal interest in a topic transparently.

Additionally, we observed replication attempts being conducted for educational purposes, either as seminar classes, theses or joined research projects. This is in line with the increasing calls to use replication studies as didactic tools (e.g. [56]). However, during the Delphi process experts perceived educational value as a secondary benefit of replication studies rather than a guiding principle of what to replicate. In other words, replication was perceived to have educational benefits regardless of which study is replicated.

### 5.2.1. Close and conceptual replications

The checklist for transparent reporting of replication target selection can be used for different types of replications. Our survey results suggested few differences in considerations between close and conceptual replications.[15] Specifically, concerns regarding generalizability were more frequently mentioned for conceptual replications, whereas motivation to avoid false-positives was more frequently mentioned for close replications. This difference is in line with the functionality of conceptual and close replications identified by Schmidt [57] and described by Zwaan *et al.* [58] as: 'Direct replications are useful for reducing false positives (i.e. claims that a specific effect exists when it was originally a chance occurrence or fluke), whereas conceptual replications provide information about the generalizability of inferences across different ways of operationally defined constructs and across different populations' ([58], p. 4).

However, we noticed many instances of a discrepancy between the label participants self-selected for their replication and the label we would have defined based on their description of the purpose of their replication. For example, respondents of close replications aimed to investigate 'a[n] specific effect with a new paradigm' (case 1) or 'the boundary conditions of phenomena' (case 12), aims that are traditionally assigned to conceptual replication [58]. Other times, following the original research protocol but changing small aspects such as outdated measures (e.g.case 32: '[we changed] nothing about the

---

[15]This might partly be due to the small proportion of RAs identifying their replication as conceptual ($n = 17$) versus close ($n = 94$), making the summary statistics for this group less stable.

procedure but we used updated and better validated measures') or imprecise measures (case 111: 'we used a different measure than originally used that gave us a more precise measure of …') resulted in respondents labelling their replication as conceptual.

Conceptual and close replications are thought to be the two ends of a continuum [59] and we indeed observed cases which situated themselves along the continuum but not at either end (e.g. 'It was somewhere in the middle of direct and conceptual', case 25). Other respondents described their replication as mixed (e.g. 'We used both' case 30) or as close with a conceptual extension (e.g. 'We combined direct replication […] and a conceptual extension […].', case 50). We did not, however, observe clear cut-off points, as for example proposed by LeBel and colleagues [60] on the continuum between close and conceptual. Minor changes (to for example the target sample or measures) were sometimes classified as close replications and other times prompted the respondent to identify their replication as conceptual.

Overall, it appears that the distinction between close and conceptual replications in practice is fuzzy at best. At times, this led to questionable scientific conduct. For example, one participant shared that while they conducted a close replication (only varying data analysis), reviewers required them to change the classification to a conceptual replication. The respondent speculated that this might have been a response 'to ease the shock of negative evidence' (case 66). Based on our data, we argue that the distinction between close and conceptual replication to be more of a theoretical than practical distinction. This possibility is given weight by the observation that distinctions between different kinds of replication vary widely in the literature [50,57]. The ambiguity in framing does not reflect the variety in kinds or replications in practice.

## 5.3. Limitations

Our results are limited by arbitrary consensus determination, that is, when do we know that consensus has been reached? This limitation is inherent to Delphi procedures (e.g. [61]). There is no agreed-upon threshold for consensus in the literature and the present use of a median of 7 with an IQR of 2 was based on previous consensus-based checklist developments (specifically [38]). However, in two instances (e.g. items regarding adaptability and pragmatism) responses were not as stable as anticipated, and whether or not an item was included hinged on the selection of responses. More precisely, the decision to in- or exclude the item changed based on which of the double responses were included in the analysis (see also the summary report for the third Delphi round). In all other instances, we observed stable ratings regardless of which responses were included. We nonetheless caution readers to perceive our checklist as complete and encourage researchers, funding agencies and other research bodies to provide feedback and recommendations. Moreover, they might want to consider adapting the checklist to their needs.

Additionally, we noted that more than half of our survey respondents came from cognitive and experimental and social psychology, potentially limiting the generalizability of our survey results. One potential explanation might be that the replication crisis in psychology rooted in social and experimental psychology (e.g. [4,5]) and calls for replications appeared earlier in social psychology making the practice more widespread in these sub-fields. Nonetheless, as our Delphi participants varied in their expertise, and as many of the concepts yielded by our analyses are applicable outside of these fields, we believe our results to generalize to most branches of social science.

## 5.4. Conclusion

Replication target selection appears to be guided by four principal factors: (1) 'doubt/uncertainty', (2) 'impact/value', (3) 'methodology/quality' and (4) 'feasibility/cost'. Replication target selection is multi-faceted and strategies for deciding what to replicate might depend on the subjective interpretation of the guiding principles. Our checklist for transparent reporting of replication target selection offers one conceptualization of these factors and prompts researchers to consider these themes when selecting a replication target. Moreover, it facilitates conversation about which studies to select for replication by providing a unified framework for how to approach and communicate such decisions.

Ethics. Ethical approval for the proposed method was granted by the Ethical Committee Psychology (ECP) of the University of Groningen, the Netherlands on 4 February 2021.

Data accessibility. Collected data, analysis code and materials are available at: https://osf.io/j7ksu/.

Supplementary material is available online [62].

Authors' contributions. M.P.: conceptualization, data curation, formal analysis, investigation, methodology, project administration, resources, visualization, writing—original draft; S.M.F., P.I. and A.E.V.: conceptualization, methodology, writing—review and editing; T.A., S.C., T.D., R.G., S.G., T.H., M.J., T.L., D.M., R.P., R.P., S.S., JM.S., E.S., US.T., M.V., J.W., N.Y. and RA.Z.: investigation; D.R.: conceptualization, formal analysis, funding acquisition, project administration, supervision, writing—review and editing.

All authors gave final approval for publication and agreed to be held accountable for the work performed therein.
Conflict of interest declaration. We declare we have no competing interest.
Funding. M.-M.P. and D.v.R. were supported by an NWO Vidi grant to D.v.R. (016.Vidi.188.001).
Acknowledgements. We would like to thank Balazs Aczel and Barnabas Szaszi for their input regarding the Delphi procedure. Additional thanks to Joyce M. Hoek for her input, guidance and reassurance on qualitative data analysis and Ymkje Anna de Vries for her regular input and supervision on the project.

# Appendix A

**Checklist for transparent reporting of replication target selection**
M.-M. Pittelkow, S. M. Field, P. M. Isager, A. E. van 't Veer, T. Anderson, S. N. Cole, T. Dominik, R. Giner-Sorolla, S. Gok, T. Heyman, M. Jekel, T. J. Luke, D. B. Mitchell, R. Peels, R. Pendrous, S. Sarrazin, J. M. Schauer, E. Specker, U. S. Tran, M. A. Vranka, J. M. Wicherts, N. Yoshimura, R. A. Zwaan & D. van Ravenzwaaij.

*Aim:* This list has been developed to assist social scientists in making informed decisions about which studies to replicate. It further serves to help them to transparently and systematically report their justifications. Importantly, this is a list of reasons why a study was actually selected, not a list of reasons why a study ought to be selected. We do not discourage or encourage a particular rationale.

*What:* This list reflects issues commonly considered when selecting a replication target. This list was developed through crowdsourcing and expert consensus. Researchers looking to select an original study for replication are advised to go over the list and use one or more considerations to transparently explain the rationale for their decision.

*When:* The list can be used by researchers that intend to replicate a study or by agencies that have to evaluate funding applications for replication studies. For the purpose of justification and decision making, we advise respondents to complete this list before the start of a replication project. For the purpose of documentation, respondents might complete this list at a later time point. However, we caution that hindsight bias might affect the accuracy of the information if it is completed after the project is complete. This list may be used to compare several targets for replication to then identify and justify the chosen replication target, or as a justification for having chosen a specific replication target after the fact.

*How:* You will be guided through a list of considerations that might apply to you. Please note that throughout the list we use the terms 'original study' and 'original claim' interchangeably . The list is divided into six sections: interest, doubt, impact, methodology and feasibility. For each section there are one or more items. For each item, you can select whether you considered the specific aspect [yes or no] and elaborate on your answer in the open text box provided. Please address both factors that applied to the original study and your plan to overcome these.

**Replication target**:
**Replication team**:

| | yes | no | n.a. |
|---|:---:|:---:|:---:|
| **With respect to your interest in the original claim, did the following factor into your considerations?** | | | |
| The relevance of the original study for your current line of research or the field you work in. | ☐ | ☐ | ☐ |
| Please elaborate: | | | |
| **With respect to possible doubt about the original claim, did the following factor into your considerations?** | | | |
| The current strength of evidence in favour of the original claim (e.g. a high/low Bayes factor, a wide/narrow confidence interval, a high/low $p$-value). | ☐ | ☐ | ☐ |
| Please elaborate: | | | |
| The (un)clarity and (un)replicability of the original protocol (e.g. completeness and clarity of the methodological description, accessibility of the materials). | ☐ | ☐ | ☐ |
| Please elaborate: | | | |

**With respect to the impact of the original study, did the following factor into your considerations?**

The importance of the original study for research (e.g. often/rarely cited, under/over-studied, published in a high/low impact journal). ☐ ☐ ☐

Please elaborate:

The theoretical relevance of the original study. ☐ ☐ ☐

Please elaborate:

Implications of the original study (e.g. for practice, policy or clinical work). ☐ ☐ ☐

Please elaborate:

**With respect to the methodology of the original study, did the following factor into your considerations?**

The sample size of the original study (too small or too large). ☐ ☐ ☐

Please elaborate:

Flaws of the original design (e.g. in- and exclusion criteria, potential confounds). ☐ ☐ ☐

Please elaborate:

Operationalization of the original study's measures (e.g. validity, reliability and bias) and how this impacts the credibility of the original study. ☐ ☐ ☐

Please elaborate:

Concern that questionable research practices have been employed (e.g. presence/absence of preregistration, potential of p-hacking or HARKing). ☐ ☐ ☐

Please elaborate:

Generalizability of the original finding (e.g. cultural and temporal context, representativeness of the sample). ☐ ☐ ☐

Please elaborate:

**With respect to the feasibility of conducting a replication of the original study, did the following factor into your considerations?**

The resources available to you for replicating the original study (e.g. funding, time, equipment, study materials or data). ☐ ☐ ☐

Please elaborate:

Your (i.e. the replicating team as a whole) presence or absence of previous experience or expertise on the original study as a practical concern. ☐ ☐ ☐

Please elaborate:

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
