## [Peer Review File · Royal Society Open Science]

Review History

RSOS-210586.R0 (Original submission)

Review form: Reviewer 1 (Massimo Grassi)

Do you have any ethical concerns with this paper?

No

Recommendation?

Reject

Comments to the Author(s)

I read the paper by Pittelkow et al and, I have to admit, I have really mixed feelings about the project. My opinion is reject (as registered report), and resubmit at the end of the Delphi process like a regular paper. However, I trust the editor and I will be happy to read a revision if the editor

thinks there is something I miss/did not understand. And of course, there is also the opinion(s) of the other reviewer(s).

Let's start from one firm point: I think the idea of the study is very interesting and may be very helpful for the community. An important tool. I also think (as far as I understand) that the procedure is sound, the paper is well written and that authors have done a very good job. But this is not the problem. Perhaps I wonder (but this is a personal opinion) whether we understood replication enough to tell something about it. Maybe in a few years time, when a quite substantial bunch of direct replications will be run.

Here copy and paste a portion of the abstract which I think includes all the core aspects of the paper I would like to discuss.

"However, a second generation of problems arises: the number of potential replication targets is at a serious mismatch with available resources. Given limited resources, replication target selection should be well-justified, systematic, and transparently communicated. At present the discussion on what to consider when selecting a replication target is limited to theoretical discussion, self-reported justifications, and a few formalized suggestions. Here, we propose a study to involve the scientific community in creating a list of considerations generally regarded important by social scientists with regards to replication target selection."

Let's start from the beginning.

[1] Do we need to submit this study as Registered Report (RR)? Perhaps I have a very restricted idea about RR but I think this wonderful tool should be used to test hypothesis. Here there is no hypothesis to test. In addition, another nice thing of RR is that authors can receive immediate feedback about their project. In this case, the editor of RSOS, and two/three reviewers (me, for example). But this project will get "naturally" much more feedback than this by all the authors that will be involved in the project. And it will receive this feedback abundantly and in several stages. I guess also that at any stage of the process (I repeat *any stage*) any of the person involved can raise his/her hand a pose a question or problem than may change radically some fundamental aspect of the project, so that the project has to start again from scratches or be drastically changed. So what's the point of asking for a RR? This type of project (Delphi type of approach) is already a sort of RR! Plus, let's assume we give IPA to the current submission and then, during the various stages, something comes out and you must change relevant portion of the process: do you violate IPA and resubmit a paper that is substantially different from the agreed one?

[2] ["the number of potential replication targets is at a serious mismatch with available resources. Given limited resources"] Perhaps it is because I work in perception, attention and cognitive science, but the first time I heard "limited resources" was ages ago. And in many case it is wrong. We simply do not know how many and how much resources we have. I just performed a search on Scopus and calculated the number of papers published in psychology in the year 2020. I excluded papers in non English language, commentaries, conference proceedings and a few other types of science-output. The net result is 58698 papers. 58698 bits of new knowledge for the field of psychology. Not too bad for a field with limited resources. And not to mention the still limited number of papers by China (4391) that produces less than half of the papers produced by Netherlands (2466). I think we have a lot of resources. We just spend them badly.

Here the Scopus search code:

```
SUBJAREA ( psyc ) AND PUBYEAR > 2019 AND PUBYEAR > 2021 AND ( LIMIT-TO (
PUBSTAGE , "final" ) ) AND ( LIMIT-TO ( DOCTYPE , "ar" ) ) AND ( EXCLUDE ( PUBYEAR ,
2021 ) ) AND ( LIMIT-TO ( LANGUAGE , "English" ) )
```

Note that if you search within these records the word "replication" you get 154 records (0.2% of the lot).

TITLE (replication) AND SUBJAREA (psyc) AND PUBYEAR < 2019 AND PUBYEAR > 2021 AND (LIMIT-TO (PUBSTAGE , "final")) AND (LIMIT-TO (DOCTYPE , "ar")) AND (EXCLUDE (PUBYEAR , 2021)) AND (LIMIT-TO (LANGUAGE , "English"))

[3] "At present the discussion on what to consider when selecting a replication target is limited to theoretical discussion, self-reported justifications, and a few formalized suggestions."

If we look at the 58698 papers above, we find that, in the largest majority of cases, we have no way to go into the creative process that generated those papers. Why do we need such a tool for replication studies? It seems to me that science runs wildly and there is not control on production (except for RR, perhaps). Why should we have/need a "speed limit" of "rules" for replications studies? The main argument seems the "lack of resources" but it looks to me weak and difficult to demonstrate. Note that we may discuss whether the "resources" are those of the publisher (e.g., journal space, server space, paper space). But (IMO), scientists should not care about these issues.

[4] "target selection should be well-justified, systematic, and transparently communicated". I think any study can be replicated in principle. Then, of course, it is a problem of the authors (and editor and reviewers, of course) to understand whether there is a potential output for the study. For example, authors do not seem to include (in the possible criteria for target selection) the replication of classic findings (to gather an precise description of the phenomenon: see Grassi et al., 2020); the replication of findings to estimate the variability and understand the modulators of a given phenomenon (Klein et al., 2018 but see also all the works by Nosek). The replication to gather an accurate estimate of the effect size of the phenomenon. Etc.

For example, in our case, we replicated a classic, found a result but it was one of the first attempt in a very large literature (the attentional blink) to provide a clear picture of the phenomenon.

Noticeably we were (as far as we know) the first to look for the N that is necessary to get a reliable picture of the phenomenon (after 30 years of studies). Note also that we were unable to replicate a specific characteristics of the original results. But reviewers said it was not important.

[5] Let's suppose that your study is now over and the results are in our pocket: what do we do with this checklist? Do you plan to suggest journals to adopt it? Else?

[6] Throughout the paper I implicitly intended "direct replication". But authors do not mention.

Massimo Grassi

References:

Klein, R. A., Vianello, M., Hasselman, F., Adams, B. G., Adams Jr, R. B., Alper, S., ... & Batra, R. (2018). Many Labs 2: Investigating variation in replicability across samples and settings.

Advances in Methods and Practices in Psychological Science, 1(4), 443-490.

Grassi, M., Crotti, C., Giofrè, D., Boedker, I., & Toffalini, E. (2020). Two replications of Raymond, Shapiro, and Arnell (1992), The Attentional Blink. Behavior Research Methods, 1-13.

Review form: Reviewer 2

Do you have any ethical concerns with this paper?

No

Recommendation?

Accept with minor revision

Comments to the Author(s)

Summary of Review:

Kudos to the authors. This was a thought-provoking and enjoyable read. I applaud the pursuit of a registered report publishing format for a project that is descriptive and exploratory. The study is well-designed and the manuscript will make a nice contribution to this interesting, important, and timely literature. None of my comments or concerns or suggestions are “deal makers” or “deal breakers” for me, but I invite the authors to consider editing the study plan and manuscript to address my comments, concerns, and questions as they see fit.

Abstract & Introduction:

I am a bit concerned about the implication that the motivations behind replication work are different from those of any work, novel or replication. From the abstract: “Given limited resources, replication target selection should be well-justified, systematic, and transparently communicated.” Is this only or particularly true of replication work? Not all work? This issue seems really interesting me, and may deserve some attention in the introduction of the paper, if not in the study design itself. I would be keen to see a direct comparison of these results to “novel” or “non-replication” work. Are the same 4 factors identified by Isager et al also driving forces for selection of research targets in a scarcity environment? A high degree of similarity seems likely to me.

It seems the most common term for psychology’s decade (and counting) of turmoil is “replication crisis,” which ties directly to your paper’s stated mission by using “replication” in the description. I’m interested in the decision not to use that term as you list descriptions of the era of uncertainty in the first sentence of the paper.

Is there empirical support/a citation for “a sharp uptick in replication studies”?

At the end of the second paragraph, perhaps you should unpack what exactly they mean by “did not replicate”?

Generally, I think the intro slightly fails to fully motivate the importance of the study. Is this work prescriptive in any way? What do we hope to learn and what impact will that have on the field?

Sampling:

Social science is a quite broad domain to plan sampling only ~30 participants from. Are you concerned that themes may differ across fields within the social sciences, and that your participants may be overrepresented by one field or underrepresented by others? You plan to ensure a balanced sample in terms of gender, career stage, and country of residence, but field/research area seem much more pertinent, and more likely to bias the themes you identify. Similarly, the initial list for possible delphi participants is made up entirely of what I might call (and consider a decent title for myself as well) “replication evangelists” . Might it be better to include potential delphi experts with more diversity in terms of replication experience and viewpoint?

Discussion:

I’d like to see a little more pre-commitment to how conclusions will be reported and discussed. The current manuscript ends rather abruptly and leaves me wondering what the final conclusions will entail.

Recommended Decision:

Again, I leave it to the authors to decide if they should change their study plan or edit their manuscript in response to my review. This project will make a contribution, and I look forward to reading future drafts of the manuscript. I recommend an in principle acceptance following minor revisions.

Review form: Reviewer 3 (Peter Branney)

Do you have any ethical concerns with this paper?

No

Recommendation?

Accept with minor revision

Comments to the Author(s)

1. The scientific validity of the research question(s).

- Concise rationale that makes an excellent case for this study. I wish the researchers the best of luck with this study.

2. The logic, rationale, and plausibility of the proposed hypotheses.

- Not applicable although (following the Mixed Methods Article Reporting Standards; <https://apastyle.apa.org/jars/mixed-table-1.pdf>) can you specify your aim or aims for this study?

3. The soundness and feasibility of the methodology and analysis pipeline (including statistical power analysis where applicable).

- Following the Mixed Methods Article Reporting Standards, what is the study design? Is this an exploratory sequential design?

- Would it also be useful to identify your population of interest/potential audience? Are you interested in all sciences or just specific disciplines (e.g. Psychology and/or cognition sciences) - the size of your population of interest should then feature in sampling and recruitment. For example, if you focused on cognition sciences, you might want to ensure you have representation across the difference components of these sciences. Or if you focus on psychology, you might want to perhaps seek representation across the range of psychological approaches. And perhaps you would only focus on disciplines/sub-disciplines in which replication is arguably of interest (e.g. Experimental approaches)

- Stage 2

- Note that you will 'evaluate whether saturation has been reached after 20-30 responses'. This is quite a range. Saturation is a contentious concept, so can you perhaps locate your use of it in the literature and clarify how it will be operationalised in this study (see e.g., doi.org/10.1080/2159676X.2019.1704846).

- Can you clarify the tense of 'choose' in Stage 2. Are you asking participants to report on their decision making (so, 'chosen', past tense)? Or are you asking for their opinion and/or attitude about ideal decision making for current (projects they are currently working on) and/or future replications? Note that at the start, you focus on people who have 'conducted' (past tense) a replication but later mention that you will include people who are conducting (so potentially present tense as they may be currently working on replication decision, although they may have already made the decision (so, past tense). I think it is important to clarify if you are looking for self-report of decision making or opinions/attitudes about ideal decision making.

- How do the Likert scale fit with qualitative survey design? How will they feature in the analysis?

○ As this is a qualitative component (although if the Likert scales are key then it will be mixed methods), can you follow the Journal Article Reporting Standards for Qualitative Methods (<https://apastyle.apa.org/jars/qual-table-1.pdf>) and include the researcher description and the researcher-participant relationship.

- Stage 3

○ Question about sample size; this seems based purely on stats rather than an underpinning theory of consensus. Surely a Delphi is akin to an election, so that any consensus is dependent on the number/proportion of the people from the relevant communities engaged in it. This seems to be a relatively unobtrusive study that should be easy to conduct at a much larger scale. If you compare this sample size to the number involved in the Open Science Collaboration, it pales in comparison.

○ Great to see you offering co-authorship for your Delphi participants. I am currently doing this (offering participants the opportunity to opt-in to co-authorship) on a study. Is it worth adding that you consider them to be 'investigators' (in semantic terms, participants are collectively helping you/each other to investigate this topic) in relation to CRediT authorship taxonomy? It might be worth clarifying if co-authorship is opt-in or a requirement of participation.

○ Personally, I think you could be much more ambitious about your sample size.

4. Whether the clarity and degree of methodological detail would be sufficient to replicate the proposed experimental procedures and analysis pipeline.

- As a consensus study, replicability is not an issue. That is, repeating this study in the future should hopefully lead to different findings. If the findings were replicated, that would unfortunately signal that open science has remained stagnant.

5. Whether the authors provide a sufficiently clear and detailed description of the methods to prevent undisclosed flexibility in the experimental procedures or analysis pipeline.

- There is disclosed flexibility in Stage 2 that is consistent with qualitative analysis.

○ There is undisclosed flexibility in how the researchers will determine if saturation has been reached in the data analysis. I would recommend clarifying how saturation will be operationalised.

- There is flexibility in the rounds of a Delphi study. That is, the researchers may decide to exclude some items and/or add new items. Can you perhaps add a section detailing how analysis will be conducted between each round? What will you do, for example, with missing data? Given the importance of flexibility, perhaps it would be useful to also add how you will document the decisions between each round (e.g. transparency log, sharing analysis code used at each round).

6. Whether the authors have considered sufficient outcome-neutral conditions (e.g. absence of floor or ceiling effects; positive controls; other quality checks) for ensuring that the results obtained are able to test the stated hypotheses.

- Not applicable for this design.

Decision letter (RSOS-210586.R0)

Dear Ms Pittelkow,

The Editors assigned to your Stage 1 Registered Report ("The Process of Replication Target Selection: What to Consider?") have now received comments from reviewers. We would like you

to revise your paper in accordance with the referee and editors suggestions which can be found below (not including confidential reports to the Editor). Please note this decision does not guarantee eventual acceptance.

Please submit a copy of your revised paper within three weeks (i.e. by the 17-Jun-2021). If we do not hear from you within this time then it will be assumed that the paper has been withdrawn. In exceptional circumstances, extensions may be possible if agreed with the Editorial Office in advance. We do not allow multiple rounds of revision so we urge you to make every effort to fully address all of the comments at this stage. If deemed necessary by the Editors, your manuscript will be sent back to one or more of the original reviewers for assessment. If the original reviewers are not available we may invite new reviewers.

When submitting your revised manuscript, you must respond to the comments made by the referees and upload a file "Response to Referees" in "Section 2 - File Upload". Please use this to document how you have responded to the comments, and the adjustments you have made. In order to expedite the processing of the revised manuscript, please be as specific as possible in your response.

Kind regards,
Professor Chris Chambers
Royal Society Open Science
openscience@royalsociety.org

on behalf of Professor Chris Chambers (Registered Reports Editor, Royal Society Open Science)
openscience@royalsociety.org

Associate Editor Comments to Author (Professor Chris Chambers):

Three expert reviewers with different disciplinary backgrounds have now reviewed the Stage 1 manuscript. The reviews are very detailed and constructive, and also quite critical in various ways. As you will see, Reviewers 2 and 3 are overall quite positive while also raising a range of issues that would need to be addressed, while Reviewer 1 is more critical and recommends outright rejection. Focusing on Reviewer 1 for moment, the reviewer's main concern is essentially three-pronged, asking why we need such a mechanism for justifying replications, whether the resource issue itself even makes sense (which essentially targets stage 1 criterion #1 of RRs at RSOS: the validity of the research question), and then even if so, whether the RR publication route is appropriate for such an open-ended initial proposal that might deviate substantially from anything that achieves IPA. Some of these critical points emerge in the other reviews -- for instance, Reviewer 2 also questions whether there is a *replication-specific* resource limit at all in science, as opposed to a general resource limit. Alongside these points, the reviewers raise issues with the robustness of the design (including sample size) and lack of detail in the qualitative methods (see Reviewer 3 especially).

All in all, this one strays close to the line for outright rejection. In a standard article format, three reviews with such extensive criticisms would almost certainly mean rejection. However, Registered Reports is a different, and often more constructive format, because it is possible in principle to address all of the concerns prior to actually running a study. In the present case, partly because of the depth and detail of the comments from reviewers, it would seem wasteful not to give you the possibility to address the issues raised. I am also motivated to offer a revision based on a helpful comment of Reviewer 1, which I actually disagree with but which helped consolidate my own thinking: that RRs are only for hypothesis driven research. I believe this is not the case, and so I am keen to see how you may be able to revise and address the reviewers' concerns for this somewhat out-of-the-box, but intriguing, submission.

Reviewer Comments to Author:

Reviewer: 1

Comments to the Author(s)

I read the paper by Pittelkow et al and, I have to admit, I have really mixed feelings about the project. My opinion is reject (as registered report), and resubmit at the end of the Delphi process like a regular paper. However, I trust the editor and I will be happy to read a revision if the editor thinks there is something I miss/did not understand. And of course, there is also the opinion(s) of the other reviewer(s).

Let's start from one firm point: I think the idea of the study is very interesting and may be very helpful for the community. An important tool. I also think (as far as I understand) that the procedure is sound, the paper is well written and that authors have done a very good job. But this is not the problem. Perhaps I wonder (but this is a personal opinion) whether we understood replication enough to tell something about it. Maybe in a few years time, when a quite substantial bunch of direct replications will be run.

Here copy and paste a portion of the abstract which I think includes all the core aspects of the paper I would like to discuss.

"However, a second generation of problems arises: the number of potential replication targets is at a serious mismatch with available resources. Given limited resources, replication target selection should be well-justified, systematic, and transparently communicated. At present the discussion on what to consider when selecting a replication target is limited to theoretical discussion, self-reported justifications, and a few formalized suggestions. Here, we propose a study to involve the scientific community in creating a list of considerations generally regarded important by social scientists with regards to replication target selection."

Let's start from the beginning.

[1] Do we need to submit this study as Registered Report (RR)? Perhaps I have a very restricted idea about RR but I think this wonderful tool should be used to test hypothesis. Here there is no hypothesis to test. In addition, another nice thing of RR is that authors can receive immediate feedback about their project. In this case, the editor of RSOS, and two/three reviewers (me, for example). But this project will get "naturally" much more feedback than this by all the authors that will be involved in the project. And it will receive this feedback abundantly and in several stages. I guess also that at any stage of the process (I repeat *any stage*) any of the person involved can raise his/her hand a pose a question or problem than may change radically some fundamental aspect of the project, so that the project has to start again from scratches or be drastically changed. So what's the point of asking for a RR? This type of project (Delphi type of approach) is already a sort of RR! Plus, let's assume we give IPA to the current submission and then, during the various stages, something comes out and you must change relevant portion of the process: do you violate IPA and resubmit a paper that is substantially different from the agreed one?

[2] ["the number of potential replication targets is at a serious mismatch with available resources. Given limited resources"] Perhaps it is because I work in perception, attention and cognitive science, but the first time I heard "limited resources" was ages ago. And in many case it is wrong. We simply do not know how many and how much resources we have. I just performed a search on Scopus and calculated the number of papers published in psychology in the year 2020. I excluded papers in non English language, commentaries, conference proceedings and a few other types of science-output. The net result is 58698 papers. 58698 bits of new knowledge for the field of psychology. Not too bad for a field with limited resources. And not to mention the still limited number of papers by China (4391) that produces less than half of the papers produced by Netherlands (2466). I think we have a lot of resources. We just spend them badly.

Here the Scopus search code:

```
SUBJAREA (psyc) AND PUBYEAR > 2019 AND PUBYEAR < 2021 AND (LIMIT-TO (PUBSTAGE, "final")) AND (LIMIT-TO (DOCTYPE, "ar")) AND (EXCLUDE (PUBYEAR, 2021)) AND (LIMIT-TO (LANGUAGE, "English"))
```

Note that if you search within these records the word "replication" you get 154 records (0.2% of the lot).

```
TITLE (replication) AND SUBJAREA (psyc) AND PUBYEAR > 2019 AND PUBYEAR < 2021 AND (LIMIT-TO (PUBSTAGE, "final")) AND (LIMIT-TO (DOCTYPE, "ar")) AND (EXCLUDE (PUBYEAR, 2021)) AND (LIMIT-TO (LANGUAGE, "English"))
```

[3] "At present the discussion on what to consider when selecting a replication target is limited to theoretical discussion, self-reported justifications, and a few formalized suggestions."

If we look at the 58698 papers above, we find that, in the largest majority of cases, we have no way to go into the creative process that generated those papers. Why do we need such a tool for replication studies? It seems to me that science runs wildly and there is not control on production (except for RR, perhaps). Why should we have/need a "speed limit" of "rules" for replications studies? The main argument seems the "lack of resources" but it looks to me weak and difficult to demonstrate. Note that we may discuss whether the "resources" are those of the publisher (e.g., journal space, server space, paper space). But (IMO), scientists should not care about these issues.

[4] "target selection should be well-justified, systematic, and transparently communicated". I think any study can be replicated in principle. Then, of course, it is a problem of the authors (and editor and reviewers, of course) to understand whether there is a potential output for the study.

For example, authors do not seem to include (in the possible criteria for target selection) the replication of classic findings (to gather an precise description of the phenomenon: see Grassi et al., 2020); the replication of findings to estimate the variability and understand the modulators of a given phenomenon (Klein et al., 2018 but see also all the works by Nosek). The replication to gather an accurate estimate of the effect size of the phenomenon. Etc.

For example, in our case, we replicated a classic, found a result but it was one of the first attempt in a very large literature (the attentional blink) to provide a clear picture of the phenomenon. Noticeably we were (as far as we know) the first to look for the N that is necessary to get a reliable picture of the phenomenon (after 30 years of studies). Note also that we were unable to replicate a specific characteristics of the original results. But reviewers said it was not important.

[5] Let's suppose that your study is now over and the results are in our pocket: what do we do with this checklist? Do you plan to suggest journals to adopt it? Else?

[6] Throughout the paper I implicitly intended "direct replication". But authors do not mention.

Massimo Grassi

References:

Klein, R. A., Vianello, M., Hasselman, F., Adams, B. G., Adams Jr, R. B., Alper, S., ... & Batra, R. (2018). Many Labs 2: Investigating variation in replicability across samples and settings. *Advances in Methods and Practices in Psychological Science*, 1(4), 443-490.

Grassi, M., Crotti, C., Giofrè, D., Boedker, I., & Toffalini, E. (2020). Two replications of Raymond, Shapiro, and Arnell (1992), The Attentional Blink. *Behavior Research Methods*, 1-13.

Reviewer: 2

Comments to the Author(s)

Summary of Review:

Kudos to the authors. This was a thought-provoking and enjoyable read. I applaud the pursuit of a registered report publishing format for a project that is descriptive and exploratory. The study is well-designed and the manuscript will make a nice contribution to this interesting, important, and timely literature. None of my comments or concerns or suggestions are “deal makers” or “deal breakers” for me, but I invite the authors to consider editing the study plan and manuscript to address my comments, concerns, and questions as they see fit.

Abstract & Introduction:

I am a bit concerned about the implication that the motivations behind replication work are different from those of any work, novel or replication. From the abstract: “Given limited resources, replication target selection should be well-justified, systematic, and transparently communicated.” Is this only or particularly true of replication work? Not all work? This issue seems really interesting me, and may deserve some attention in the introduction of the paper, if not in the study design itself. I would be keen to see a direct comparison of these results to “novel” or “non-replication” work. Are the same 4 factors identified by Isager et al also driving forces for selection of research targets in a scarcity environment? A high degree of similarity seems likely to me.

It seems the most common term for psychology’s decade (and counting  ) of turmoil is “replication crisis,” which ties directly to your paper’s stated mission by using “replication” in the description. I’m interested in the decision not to use that term as you list descriptions of the era of uncertainty in the first sentence of the paper.

Is there empirical support/a citation for “a sharp uptick in replication studies”?

At the end of the second paragraph, perhaps you should unpack what exactly they mean by “did not replicate”?

Generally, I think the intro slightly fails to fully motivate the importance of the study. Is this work prescriptive in any way? What do we hope to learn and what impact will that have on the field?

Sampling:

Social science is a quite broad domain to plan sampling only ~30 participants from. Are you concerned that themes may differ across fields within the social sciences, and that your participants may be overrepresented by one field or underrepresented by others? You plan to ensure a balanced sample in terms of gender, career stage, and country of residence, but field/research area seem much more pertinent, and more likely to bias the themes you identify. Similarly, the initial list for possible delphi participants is made up entirely of what I might call

(and consider a decent title for myself as well) “replication evangelists” ♦♦. Might it be better to include potential delphi experts with more diversity in terms of replication experience and viewpoint?

Discussion:

I'd like to see a little more pre-commitment to how conclusions will be reported and discussed. The current manuscript ends rather abruptly and leaves me wondering what the final conclusions will entail.

Recommended Decision:

Again, I leave it to the authors to decide if they should change their study plan or edit their manuscript in response to my review. This project will make a contribution, and I look forward to reading future drafts of the manuscript. I recommend an in principle acceptance following minor revisions.

Reviewer: 3

Comments to the Author(s)

1. The scientific validity of the research question(s).

- Concise rationale that makes an excellent case for this study. I wish the researchers the best of luck with this study.

2. The logic, rationale, and plausibility of the proposed hypotheses.

- Not applicable although (following the Mixed Methods Article Reporting Standards; <https://apastyle.apa.org/jars/mixed-table-1.pdf>) can you specify your aim or aims for this study?

3. The soundness and feasibility of the methodology and analysis pipeline (including statistical power analysis where applicable).

- Following the Mixed Methods Article Reporting Standards, what is the study design? Is this an exploratory sequential design?

- Would it also be useful to identify your population of interest/potential audience? Are you interested in all sciences or just specific disciplines (e.g. Psychology and/or cognition sciences) - the size of your population of interest should then feature in sampling and recruitment. For example, if you focused on cognition sciences, you might want to ensure you have representation across the difference components of these sciences. Or if you focus on psychology, you might want to perhaps seek representation across the range of psychological approaches. And perhaps you would only focus on disciplines/sub-disciplines in which replication is arguably of interest (e.g. Experimental approaches)

- Stage 2

- Note that you will 'evaluate whether saturation has been reached after 20-30 responses'. This is quite a range. Saturation is a contentious concept, so can you perhaps locate your use of it in the literature and clarify how it will be operationalised in this study (see e.g., doi.org/10.1080/2159676X.2019.1704846).

- Can you clarify the tense of 'choose' in Stage 2. Are you asking participants to report on their decision making (so, 'chosen', past tense)? Or are you asking for their opinion and/or attitude about ideal decision making for current (projects they are currently working on) and/or future replications? Note that at the start, you focus on people who have 'conducted' (past tense) a replication but later mention that you will include people who are conducting (so potentially present tense as they may be currently working on replication decision, although they may have already made the decision (so, past tense). I think it is important to clarify if you are looking for self-report of decision making or opinions/attitudes about ideal decision making.

- How do the Likert scale fit with qualitative survey design? How will they feature in the analysis?
- As this is a qualitative component (although if the Likert scales are key then it will be mixed methods), can you follow the Journal Article Reporting Standards for Qualitative Methods (<https://apastyle.apa.org/jars/qual-table-1.pdf>) and include the researcher description and the researcher-participant relationship.
 - Stage 3
 - Question about sample size; this seems based purely on stats rather than an underpinning theory of consensus. Surely a Delphi is akin to an election, so that any consensus is dependent on the number/proportion of the people from the relevant communities engaged in it. This seems to be a relatively unobtrusive study that should be easy to conduct at a much larger scale. If you compare this sample size to the number involved in the Open Science Collaboration, it pales in comparison.
 - Great to see you offering co-authorship for your Delphi participants. I am currently doing this (offering participants the opportunity to opt-in to co-authorship) on a study. Is it worth adding that you consider them to be 'investigators' (in semantic terms, participants are collectively helping you/each other to investigate this topic) in relation to CRediT authorship taxonomy? It might be worth clarifying if co-authorship is opt-in or a requirement of participation.
 - Personally, I think you could be much more ambitious about your sample size.
- 4. Whether the clarity and degree of methodological detail would be sufficient to replicate the proposed experimental procedures and analysis pipeline.
 - As a consensus study, replicability is not an issue. That is, repeating this study in the future should hopefully lead to different findings. If the findings were replicated, that would unfortunately signal that open science has remained stagnant.
- 5. Whether the authors provide a sufficiently clear and detailed description of the methods to prevent undisclosed flexibility in the experimental procedures or analysis pipeline.
 - There is disclosed flexibility in Stage 2 that is consistent with qualitative analysis.
 - There is undisclosed flexibility in how the researchers will determine if saturation has been reached in the data analysis. I would recommend clarifying how saturation will be operationalised.
 - There is flexibility in the rounds of a Delphi study. That is, the researchers may decide to exclude some items and/or add new items. Can you perhaps add a section detailing how analysis will be conducted between each round? What will you do, for example, with missing data? Given the importance of flexibility, perhaps it would be useful to also add how you will document the decisions between each round (e.g. transparency log, sharing analysis code used at each round).
- 6. Whether the authors have considered sufficient outcome-neutral conditions (e.g. absence of floor or ceiling effects; positive controls; other quality checks) for ensuring that the results obtained are able to test the stated hypotheses.
 - Not applicable for this design.

Author's Response to Decision Letter for (RSOS-210586.R0)

See Appendix A.

RSOS-210586.R0 (Revision)

Review form: Reviewer 1 (Massimo Grassi)

Do you have any ethical concerns with this paper?

No

Recommendation?

Accept in principle

Comments to the Author(s)

Ok for me. I thank the authors for the extensive replies. I would just change "close replication" with "direct replication" which is, AFAIK, the current term used in literature. Of course, feel free to drop this suggestion if I'm wrong.

Regards,

M

Review form: Reviewer 2

Do you have any ethical concerns with this paper?

No

Recommendation?

Accept in principle

Comments to the Author(s)

The authors have adequately addressed my concerns and incorporated many of my suggestions into improvements to the study plan and the paper itself. I recommend in principle acceptance at this stage and look forward to reviewing future iterations of this manuscript!

Decision letter (RSOS-210586.R1)

Dear Ms Pittelkow

On behalf of the Editor, I am pleased to inform you that your Manuscript RSOS-210586.R1 entitled "The Process of Replication Target Selection in Psychology: What to Consider?" has been accepted in principle for publication in Royal Society Open Science. The reviewers' and editors' comments are included at the end of this email.

You may now progress to Stage 2 and complete the study as approved. Before commencing data collection we ask that you:

1) Update the journal office as to the anticipated completion date of your study.

2) Register your approved protocol on the Open Science Framework (<https://osf.io/>) or other recognised repository, either publicly or privately under embargo until submission of the Stage 2 manuscript. Please note that a time-stamped, independent registration of the protocol is mandatory under journal policy, and manuscripts that do not conform to this requirement cannot be considered at Stage 2. The protocol should be registered unchanged from its current approved state, with the time-stamp preceding implementation of the approved study design. We strongly recommend using the dedicated registration portal for accepted Stage 1 Registered Reports hosted by the OSF at <https://osf.io/rr>

Following completion of your study, we invite you to resubmit your paper for peer review as a Stage 2 Registered Report. Please note that your manuscript can still be rejected for publication at Stage 2 if the Editors consider any of the following conditions to be met:

- The results were unable to test the authors' proposed hypotheses by failing to meet the approved outcome-neutral criteria.
- The authors altered the Introduction, rationale, or hypotheses, as approved in the Stage 1 submission.
- The authors failed to adhere closely to the registered experimental procedures. Please note that any deviations from the approved experimental procedures must be communicated to the editor immediately for approval, and prior to the completion of data collection. Failure to do so can result in revocation of in-principle acceptance and rejection at Stage 2 (see complete guidelines for further information).
- Any post-hoc (unregistered) analyses were either unjustified, insufficiently caveated, or overly dominant in shaping the authors' conclusions.
- The authors' conclusions were not justified given the data obtained.

We encourage you to read the complete guidelines for authors concerning Stage 2 submissions at <https://royalsocietypublishing.org/rsos/registered-reports#ReviewerGuideRegRep>. Please especially note the requirements for data sharing, reporting the URL of the independently registered protocol, and that withdrawing your manuscript will result in publication of a Withdrawn Registration.

Once again, thank you for submitting your manuscript to Royal Society Open Science and we look forward to receiving your Stage 2 submission. If you have any questions at all, please do not hesitate to get in touch. We look forward to hearing from you shortly with the anticipated submission date for your stage two manuscript.

on behalf of Professor Chris Chambers (Registered Reports Editor, Royal Society Open Science)
openscience@royalsociety.org

Associate Editor Comments to Author (Professor Chris Chambers):

Associate Editor: 1

Comments to the Author:

Two of the three reviewers were available to review the revised manuscript, and I'm happy to say that both are now satisfied and recommend IPA. This was a very substantial revision that strayed close to the rejection line, and so the authors have done very well to turn it around and

achieve IPA so readily. Reviewer 1 (who was most critical in the first round) offers a final suggestion concerning terminology. I am happy for the authors to use either term, as I have seen both used in various settings -- the main thing is to be consistent and if making this change please do so before formally registering the Stage 1 manuscript (which you need to do prior to starting the study). For now, good luck and we look forward to seeing it again at Stage 2.

Reviewers' comments to Author:

Reviewer: 1

Comments to the Author(s)

Ok for me. I thank the authors for the extensive replies. I would just change "close replication" with "direct replication" which is, AFAIK, the current term used in literature. Of course, feel free to drop this suggestion if I'm wrong.

Regards,

m

Reviewer: 2

Comments to the Author(s)

The authors have adequately addressed my concerns and incorporated many of my suggestions into improvements to the study plan and the paper itself. I recommend in principle acceptance at this stage and look forward to reviewing future iterations of this manuscript!

Author's Response to Decision Letter for (RSOS-210586.R1)

See Appendix B.

RSOS-210586.R2

Review form: Reviewer 1 (Massimo Grassi)

Is the manuscript scientifically sound in its present form?

Yes

Are the interpretations and conclusions justified by the results?

Yes

Is the language acceptable?

Yes

Do you have any ethical concerns with this paper?

No

Have you any concerns about statistical analyses in this paper?

No

Recommendation?

Accept with minor revision

Comments to the Author(s)

I read the stage 2 and my opinion oscillates between "accept as is" and "minor revisions". The main problem I see now in the paper is length. Although the final paper is super transparent and reports everything, I think it would be more important for the community if it was shorter. My 2 cents suggestion is to move part of it (ie the process) somewhere else and keep beginning and end.

Minors.

1. I saw that in the "The present study" section there are still a few future tense. I saw authors removed them from the rest of the paper. To me, that part looks it should be written in past tense. But I'm not an English speaker therefore please drop this point no sense.
2. I found captions a bit too short and with few details. In such a long paper, figure, tables (and captions) are vital, I think. Figure 3 in particular is there with no much explanation.

Very good job.

Massimo Grassi

Review form: Reviewer 3 (Peter Branney)

Is the manuscript scientifically sound in its present form?

Yes

Are the interpretations and conclusions justified by the results?

Yes

Is the language acceptable?

Yes

Do you have any ethical concerns with this paper?

No

Have you any concerns about statistical analyses in this paper?

No

Recommendation?

Accept with minor revision

Comments to the Author(s)

This is my second time reviewing a Stage 2 submission, so I am being guided (and will use subheadings to emphasise) the five key issues to consider Royal Society Open Science, Registered Report, 'Reviewer Guidelines' (<https://royalsocietypublishing.org/rsos/registered-reports>). Overall, the paper is easy to read and provides sufficient detail to understand what they did and why, and how they came to the conclusions in creating a transparent checklist.

1) Whether the data are able to test the authors' proposed hypotheses by passing the approved outcome-neutral criteria (such as absence of floor and ceiling effects or success of positive controls or other quality checks). Failure to pass these conditions may lead to manuscript rejection.

Yes. The data are in a folder called 'supplementary files' on the OSF project page and there is a sub folder for all three stages of the Delphi Exercise. In Stage 2, for example, the survey responses are complemented by the qualitative codebook and the reflexive notes. This really helps show the process of the of data analysis. In the manuscript, is it worth specifying what license you have given the data, so that readers can be clear about what they can, and cannot, do with it?

In looking at the data, I am wondering if it meets the FAIR principles for data stewardship (see Step 3 in DOI: 10.31234/osf.io/ahdcu). I wonder if using the term 'supplementary file' might limit 'accessibility' (rather than say, the 'data' component in OSF). My understanding is that 'supplementary' is in reference to this manuscript but the OSF project page is for a study, so this reference point might be unclear. When I looked at the project page, why first question was 'where is the data', and it took me some time to figure it out. Also, if you add the license, you'll improve the 'reusability' (and a data component can have a different license to the project folder). Could you also include copies of the information and consent participants consented to - consent it something I've found is incredibly important in interviewing qualitative researchers about open data (DOI:10/gf429z)? For anyone wanting to look at the data, they may need to see precisely what participants consented to before deciding if they can or should use it. I also wonder if the OSF is the best home for research data. Will this data, for example, be 'discoverable' for researchers using Internet search? Would a specialist data archive, such as the UK Data Service or the Finnish Social Science Data Archive, be a better home or 'steward' for your data, particularly over the long-term? Last, the paper embeds URL to relevant places in the OSF project page, although I would recommend following the journal style and citing them. If the urls ever become broken, the additional information in the citation should help readers find it.

2) Whether the Introduction, rationale and stated hypotheses are the same as the approved Stage1 submission (required)

Yes, as far as I can tell, they are the same.

3) Whether the authors adhered precisely to the registered experimental procedures

There were some deviations, which are described and justified. In relation the qualitative analysis, the explanation of the deviation is helpful in clarifying the role of the researchers in interpreting the data. I also think explanation of these deviations helps explain the decision in the Discussion section for a checklist for transparent reporting.

4) Where applicable, whether any unregistered exploratory statistical analyses are justified, methodologically sound, and informative.

Not applicable.

5) Whether the authors' conclusions are justified given the data. Please note that editorial decisions will not be based on the perceived importance, novelty, or conclusiveness of the results.

Yes. I look forward to seeing the use of this checklist in future research. For the avoidance of any doubt, can you include the final checklist/a reference to the final checklist. E.g. "The final checklist (reference)..." p. 24, first line/line 892.

Typographical

Is the citation on page four, first line, correct "a consensus-based method1". Looking at the title of the paper cited, it doesn't look relevant.

Peter Branney

Decision letter (RSOS-210586.R2)

Dear Ms Pittelkow:

On behalf of the Editor, I am pleased to inform you that your Stage 2 Registered Report RSOS-210586.R2 entitled "The Process of Replication Target Selection in Psychology: What to Consider?" has been deemed suitable for publication in Royal Society Open Science subject to minor revision in accordance with the referee suggestions. Please find the referees' comments at the end of this email.

The reviewers and Subject Editor have recommended publication, but also suggest some minor revisions to your manuscript. We invite you to respond to the comments and revise your manuscript. Below the referees' and Editors' comments (where applicable) we provide additional requirements. Final acceptance of your manuscript is dependent on these requirements being met. We provide guidance below to help you prepare your revision.

Please submit your revised manuscript and required files (see below) no later than 7 days from today's (ie 03-Jan-2023) date. Note: the ScholarOne system will 'lock' if submission of the revision is attempted 7 or more days after the deadline. If you do not think you will be able to meet this deadline please contact the editorial office immediately.

on behalf of Professor Chris Chambers
(Registered Reports Editor, Royal Society Open Science)
openscience@royalsociety.org

Associate Editor Comments to Author (Professor Chris Chambers):

Associate Editor: 1

Comments to the Author:

I have now obtained evaluations of your submission from two of the reviewers who assessed your manuscript at Stage 1. Both are positive, as is my own assessment. The reviews nevertheless contain some helpful reflections and recommendations for minor improvements. Concerning the suggestion by Reviewer 1 to shorten the manuscript, and I felt the length was acceptable as the manuscript is well organised and it doesn't feel overly long. However, I am happy for you to shorten it if you prefer. If you do so, please focus any such reduction on results and discussion; please do not remove any content that was approved at Stage 1.

Concerning Reviewer 3's comment about using the OSF, this is an acceptable repository from the journal's point of view.

I look forward to receiving your revised submission and response to the reviewers, after which I expect to be able to issue full acceptance without further in-depth review.

Comments to Author:

Reviewer: 1

Comments to the Author(s)

I read the stage 2 and my opinion oscillates between "accept as is" and "minor revisions". The main problem I see now in the paper is length. Although the final paper is super transparent and reports everything, I think it would be more important for the community if it was shorter. My 2 cents suggestion is to move part of it (ie the process) somewhere else and keep beginning and end.

Minors.

1. I saw that in the "The present study" section there are still a few future tense. I saw authors removed them from the rest of the paper. To me, that part looks it should be written in past tense. But I'm not an English speaker therefore please drop this point no sense.
2. I found captions a bit too short and with few details. In such a long paper, figure, tables (and captions) are vital, I think. Figure 3 in particular is there with no much explanation.

Very good job.

Massimo Grassi

Reviewer: 3

Comments to the Author(s)

This is my second time reviewing a Stage 2 submission, so I am being guided (and will use subheadings to emphasise) the five key issues to consider Royal Society Open Science, Registered Report, 'Reviewer Guidelines' (<https://royalsocietypublishing.org/rsos/registered-reports>). Overall, the paper is easy to read and provides sufficient detail to understand what they did and why, and how they came to the conclusions in creating a transparent checklist.

- 1) Whether the data are able to test the authors' proposed hypotheses by passing the approved outcome-neutral criteria (such as absence of floor and ceiling effects or success of positive controls or other quality checks). Failure to pass these conditions may lead to manuscript rejection.

Yes. The data are in a folder called 'supplementary files' on the OSF project page and there is a sub folder for all three stages of the Delphi Exercise. In Stage 2, for example, the survey responses are complemented by the qualitative codebook and the reflexive notes. This really helps show the process of the of data analysis. In the manuscript, is it worth specifying what license you have given the data, so that readers can be clear about what they can, and cannot, do with it?

In looking at the data, I am wondering if it meets the FAIR principles for data stewardship (see Step 3 in DOI: 10.31234/osf.io/ahdcu). I wonder if using the term 'supplementary file' might limit 'accessibility' (rather than say, the 'data' component in OSF). My understanding is that 'supplementary' is in reference to this manuscript but the OSF project page is for a study, so this reference point might be unclear. When I looked at the project page, why first question was 'where is the data', and it took me some time to figure it out. Also, if you add the license, you'll improve the 'reusability' (and a data component can have a different license to the project folder). Could you also include copies of the information and consent participants consented to - consent

it something I've found is incredibly important in interviewing qualitative researchers about open data (DOI:10/gf429z)? For anyone wanting to look at the data, they may need to see precisely what participants consented to before deciding if they can or should use it. I also wonder if the OSF is the best home for research data. Will this data, for example, be 'discoverable' for researchers using Internet search? Would a specialist data archive, such as the UK Data Service or the Finnish Social Science Data Archive, be a better home or 'steward' for your data, particularly over the long-term? Last, the paper embeds URL to relevant places in the OSF project page, although I would recommend following the journal style and citing them. If the urls ever become broken, the additional information in the citation should help readers find it.

2) Whether the Introduction, rationale and stated hypotheses are the same as the approved Stage1 submission (required)

Yes, as far as I can tell, they are the same.

3) Whether the authors adhered precisely to the registered experimental procedures

There were some deviations, which are described and justified. In relation the qualitative analysis, the explanation of the deviation is helpful in clarifying the role of the researchers in interpreting the data. I also think explanation of these deviations helps explain the decision in the Discussion section for a checklist for transparent reporting.

4) Where applicable, whether any unregistered exploratory statistical analyses are justified, methodologically sound, and informative.

Not applicable.

5) Whether the authors' conclusions are justified given the data. Please note that editorial decisions will not be based on the perceived importance, novelty, or conclusiveness of the results.

Yes. I look forward to seeing the use of this checklist in future research. For the avoidance of any doubt, can you include the final checklist/a reference to the final checklist. E.g. "The final checklist (reference)..." p. 24, first line/line 892.

Typographical

Is the citation on page four, first line, correct "a consensus-based method1". Looking at the title of the paper cited, it doesn't look relevant.

Peter Branney

===PREPARING YOUR MANUSCRIPT===

one version should clearly identify all the changes that have been made (for instance, in coloured highlight, in bold text, or tracked changes);

===PREPARING YOUR REVISION IN SCHOLARONE===

- If you are providing image files for potential cover images, please upload these at this step, and inform the editorial office you have done so. You must hold the copyright to any image provided.
- A copy of your point-by-point response to referees and Editors. This will expedite the preparation of your proof.

- Ensure that your data access statement meets the requirements at <https://royalsociety.org/journals/authors/author-guidelines/#data>. You should ensure that you cite the dataset in your reference list. If you have deposited data etc in the Dryad repository, please only include the 'For publication' link at this stage. You should remove the 'For review' link.
- If you are requesting an article processing charge waiver, you must select the relevant waiver option (if requesting a discretionary waiver, the form should have been uploaded, see 'File upload' above).
- If you have uploaded any electronic supplementary (ESM) files, please ensure you follow the guidance at <https://royalsociety.org/journals/authors/author-guidelines/#supplementary-material> to include a suitable title and informative caption. An example of appropriate titling and captioning may be found at https://figshare.com/articles/Table_S2_from_Is_there_a_trade-off_between_peak_performance_and_performance_breadth_across_temperatures_for_aerobic_scope_in_teleost_fishes_/3843624.

Author's Response to Decision Letter for (RSOS-210586.R2)

See Appendix C.

Decision letter (RSOS-210586.R3)

Dear Ms Pittelkow:

I am pleased to inform you that your manuscript entitled "The Process of Replication Target Selection in Psychology: What to Consider?" is now accepted for publication in Royal Society Open Science.

Please remember to make any data sets or code libraries 'live' prior to publication, and update any links as needed when you receive a proof to check - for instance, from a private 'for review' URL to a publicly accessible 'for publication' URL. It is also good practice to add data sets, code and other digital materials to your reference list.

Royal Society Open Science is a fully open access journal. A payment may be due before your article is published. The Royal Society has partnered with Copyright Clearance Center's (CCC's) RightsLink service to allow authors to pay article processing charges or page charges. After your manuscript has been accepted, the corresponding author will receive an email from CCC with the subject "Please submit your article processing/open access charge(s)/page charges" inviting you to pay your charges or request an invoice. The email from CCC will come from the email domain @copyright.com (if you have any queries regarding fees, please see <https://royalsocietypublishing.org/rsos/charges> or contact authorfees@royalsociety.org). If you request an invoice, it will be sent to you from CCC. It is important to be cautious about payment scams. If you receive an email or text message requesting payment and have any concerns, we recommend contacting us through our website, rather than clicking on any links. **The Society will never ask you to make a direct payment.**

Follow Royal Society Publishing on Twitter: @RSocPublishing
Follow Royal Society Publishing on Facebook:
<https://www.facebook.com/RoyalSocietyPublishing/>
Read Royal Society Publishing's blog:
<https://royalsociety.org/blog/blogsearchpage/?category=Publishing>

Appendix A

Also on behalf of my co-authors, I would like to thank the reviewers for their insightful comments and suggestions. We found the reviews very constructive and are convinced that they helped to significantly improve our manuscript and planning.

Please find our point by point responses below. Our replies are printed in bold and line numbers refer to the manuscript file including track changes.

Kind regards,

Merle-Marie Pittelkow

Reviewer: 1

Comments to the Author(s)

I read the paper by Pittelkow et al and, I have to admit, I have really mixed feelings about the project. My opinion is reject (as registered report), and resubmit at the end of the Delphi process like a regular paper. However, I trust the editor and I will be happy to read a revision if the editor thinks there is something I miss/did not understand. And of course, there is also the opinion(s) of the other reviewer(s).

Let's start from one firm point: I think the idea of the study is very interesting and may be very helpful for the community. An important tool. I also think (as far as I understand) that the procedure is sound, the paper is well written and that authors have done a very good job. But this is not the problem. Perhaps I wonder (but this is a personal opinion) whether we understood replication enough to tell something about it. Maybe in a few years time, when a quite substantial bunch of direct replications will be run.

Here copy and paste a portion of the abstract which I think includes all the core aspects of the paper I would like to discuss.

"However, a second generation of problems arises: the number of potential replication targets is at a serious mismatch with available resources. Given limited resources, replication target selection should be well-justified, systematic, and transparently communicated. At present the discussion on what to consider when selecting a replication target is limited to theoretical discussion, self-reported justifications, and a few formalized suggestions. Here, we propose a study to involve the scientific community in creating a list of considerations generally regarded important by social scientists with regards to replication target selection."

Let's start from the beginning.

[1] Do we need to submit this study as Registered Report (RR)? Perhaps I have a very restricted idea about RR but I think this wonderful tool should be used to test hypothesis. Here there is no hypothesis to test. In addition, another nice thing of RR is that authors can receive immediate feedback about their project. In this case, the editor of RSOS, and two/three reviewers (me, for example). But this project will get "naturally" much more feedback than this by all the authors that will be involved in the project. And it will receive this

feedback abundantly and in several stages. I guess also that at any stage of the process (I repeat *any stage*) any of the person involved can raise his/her hand a pose a question or problem than may change radically some fundamental aspect of the project, so that the project has to start again from scratches or be drastically changed. So what's the point of asking for a RR? This type of project (Delphi type of approach) is already a sort of RR! Plus, let's assume we give IPA to the current submission and then, during the various stages, something comes out and you must change relevant portion of the process: do you violate IPA and resubmit a paper that is substantially different from the agreed one?

We recognize the reviewer's concern. However, in our opinion the arguments provided by the reviewer support an RR submission all the more. The reviewer raises the concern that no explicit hypotheses make RR an unsuitable submission format for our proposal. Although it is true that our proposal features no predefined hypotheses, our manuscript still benefits from the critique provided in stage 1 for our study's aims, design, sampling plans, plans for the findings once they are in, and for the intended use of the overall tool once it is finished. Hypotheses are just one piece in the bigger picture that is the research process, and we believe the RR process has the potential to boost the quality, validity and impact of all studies that follow the typical research pipeline. Moreover, RRs not only serve to constrain analytic flexibility, but also (and perhaps more importantly) hinder result-based publication bias by preventing journals from selecting on results. We therefore think that any empirical study would benefit from being submitted as a RR, regardless of study design.

While it is true that the project will get input during both the survey and the Delphi procedure, this input pertains to the potential results of the study (i.e., the list of considerations) and not the design per se. We respectfully disagree with the reviewer that the project will 'naturally' receive input regarding its design and set-up. Even if a problem would arise that fundamentally changes the nature of the design, we believe that this is not a reason not to conduct this project as an RR. After all, similar problems apply equally to more traditional confirmatory RR designs, where unforeseen circumstances call for transparently documented changes in the design -- for example having to switch from collecting participant data face-to-face to alternative online sampling.

As for having to request deviation from IPA in case of changes, we see this as an opportunity to critically reflect on the need for potential changes and a chance to transparently report the evolution of the project. The RR submission format is still relatively novel, and we believe that cases such as ours provide journals with the opportunity to explore and demonstrate the benefits of the RR process to non-confirmatory studies. Although scientific quality and integrity are safeguarded by registration of study plans and the restriction of some activities, science is a creative process and we believe excluding exploratory and qualitative research from the RR submission format would hinder scientific progress.

[2] ["the number of potential replication targets is at a serious mismatch with available resources. Given limited resources"] Perhaps it is because I work in perception, attention and cognitive science, but the first time I heard "limited resources" was ages ago. And in many case it is wrong. We simply do not know how many and how much resources we have. I just performed a search on Scopus and calculated the number of

papers published in psychology in the year 2020. I excluded papers in non English language, commentaries, conference proceedings and a few other types of science-output. The net result is 58698 papers. 58698 bits of new knowledge for the field of psychology. Not too bad for a field with limited resources. And not to mention the still limited number of papers by China (4391) that produces less than half of the papers produced by Netherlands (2466). I think we have a lot of resources. We just spend them badly.

Here the Scopus search code:

```
SUBJAREA ( psyc ) AND PUBYEAR > 2019 AND PUBYEAR < 2021 AND ( LIMIT-TO ( PUBSTAGE , "final" ) ) AND ( LIMIT-TO ( DOCTYPE , "ar" ) ) AND ( EXCLUDE ( PUBYEAR , 2021 ) ) AND ( LIMIT-TO ( LANGUAGE , "English" ) )
```

Note that if you search within these records the word "replication" you get 154 records (0.2% of the lot).

```
TITLE ( replication ) AND SUBJAREA ( psyc ) AND PUBYEAR > 2019 AND PUBYEAR < 2021 AND ( LIMIT-TO ( PUBSTAGE , "final" ) ) AND ( LIMIT-TO ( DOCTYPE , "ar" ) ) AND ( EXCLUDE ( PUBYEAR , 2021 ) ) AND ( LIMIT-TO ( LANGUAGE , "English" ) )
```

While the body of scientific literature and thus the number of potential replication targets is ever-growing, the resources for such studies remain limited. We do not want to argue that the sum of all resources in (in this case) psychology combined is limited. Instead, we argue that the resources available to any one researcher at a given point in time is limited compared with the number of studies they might be interested in replicating, resulting in only a fragment of scientific output being replications. We agree that we should spend our resources wisely. Currently, researchers are constantly in a position where they must choose between several potential replication studies. We aim to make the thought process behind such decisions more transparent.

From the granting agencies side, the argument points in the same direction: The few funding agencies that do provide grants for replication studies specifically receive many good proposals, but can only fund a low percentage of them. For instance, the last call from the Dutch Research Council (NWO) for replication studies specifically funded only 7 out of 41 proposals. From their point of view, having more transparent justifications for conducting certain replication studies arguably leads to more effective allocation of resources.

To make our point clearer, we adopted the following changes:

- **Starting line 30 ff. :**

Increased interest in the discussion and execution of replication studies contributes to the active effort to restore credibility to scientific research, *including psychological research*. However, it brings with it a second generation of problems. Among these is the fact that the number of potential replication targets is at a serious mismatch with *the resources available for replication studies*, both in terms of human labour and in terms of available funds. *As one example, in a separate project author AvtV and PMI aim to replicate original research in social neuroscience [10]. Even restricting their candidate set to studies using fMRI in the last ten*

years, they currently have a pool of over two thousand potential targets to select from. The rate at which empirical studies in psychology are published has been growing exponentially for the past century. Simultaneously, the rate at which original studies are replicated is very low. The replication rate in social sciences and psychology alike has been estimated at around 1% [8,11], though the rate is difficult to estimate exactly. While the pile of potential replication targets is growing at an exponential rate, funding for replication is developing more slowly. This results in an enormous back-log of non-replicated research to contend with.

To accommodate the need for replication studies, funding opportunities targeting replication that have emerged range from broad scale funding opportunities in the Biomedical Sciences [e.g., 12], Social Sciences and Humanities [e.g.,9,13,14], or Educational Sciences [e.g.,15], to specific initiatives calling for replication in pre-specified areas [e.g.,16]. *Even so, grants for replications receive many good proposals, but can only fund a low percentage of them. For example, the Dutch funder NWO could only fund around 10% of submitted replication studies [17].* Though there is an increase in the number of funding opportunities, they remain relatively scarce and overall resources for replication studies remain limited.

[3] "At present the discussion on what to consider when selecting a replication target is limited to theoretical discussion, self-reported justifications, and a few formalized suggestions."

If we look at the 58698 papers above, we find that, in the largest majority of cases, we have no way to go into the creative process that generated those papers. Why do we need such a tool for replication studies? It seems to me that science runs wildly and there is not control on production (except for RR, perhaps). Why should we have/need a "speed limit" of "rules" for replications studies? The main argument seems the "lack of resources" but it looks to me weak and difficult to demonstrate. Note that we may discuss whether the "resources" are those of the publisher (e.g., journal space, server space, paper space). But (IMO), scientists should not care about these issues.

It seems that we are in disagreement with the reviewer on this point. We believe that transparent reporting of the rationale for conducting *any* study, regardless of the study design, is important for ensuring responsible spending of the taxpayers money. Moreover, we do not perceive systematic reporting of the reasoning behind study selection as limiting the scientific process. To be clear, we are not prescribing certain standards for when a study should be replicated but are interested in (1) describing how researchers made this decision in the past and (2) evaluate whether there is consensus on what aspects to consider when selecting a replication target. At present, replication as well as it's selection is poorly understood and our survey will shed more light on what kind of replications are done for what reason. The Delphi process afterwards, will reveal considerations important to psychological researchers in general when selecting a replication target. These should by no means hinder the creative process of science but simply enable researchers to make it transparent and visible to others (i.e., other researchers, funding agencies, the public, etc.).

We added some clarifying sentences in the manuscript:

- Starting line 64 ff. :

Whatever the reasons for selecting a particular replication target, we believe that communicating how the eventual decision was reached is very important. *At present, there is no consensus as to what characterizes a study "worth replicating" or "in need of replication". Regardless of whether or not consensus on this matter can possibly be achieved, clearly communicating one's reasoning behind selecting a replication target enables others to understand, and evaluate the decision. To spend limited resources for replication studies wisely, it is in the interest of both researchers and funding agencies to replicate studies that make sense and that make good use of the resources. Having a transparent logbook of why targets are selected for replication is a first step towards spending limited resources well.*

[4] "target selection should be well-justified, systematic, and transparently communicated". I think any study can be replicated in principle. Then, of course, it is a problem of the authors (and editor and reviewers, of course) to understand whether there is a potential output for the study. For example, authors do not seem to include (in the possible criteria for target selection) the replication of classic findings (to gather an precise description of the phenomenon: see Grassi et al., 2020); the replication of findings to estimate the variability and understand the modulators of a given phenomenon (Klein et al., 2018 but see also all the works by Nosek). The replication to gather an accurate estimate of the effect size of the phenomenon. Etc.

For example, in our case, we replicated a classic, found a result but it was one of the first attempt in a very large literature (the attentional blink) to provide a clear picture of the phenomenon. Noticeably we were (as far as we know) the first to look for the N that is necessary to get a reliable picture of the phenomenon (after 30 years of studies). Note also that we were unable to replicate a specific characteristics of the original results. But reviewers said it was not important.

We agree with the reviewer that any study can be replicated in principle. As a result, any given researcher has many potential studies they could, in principle, replicate. As the researcher is constrained by resources, they must choose one or more replication targets from the large pool of potential targets. How should such a choice be made? Given that the researcher has a certain goal they want to achieve with their replication (e.g., minimize uncertainty about the finding), and given that the choice of which study to replicate affects the researcher's ability to reach their goal (some studies are highly likely to replicate; for other studies the replication result is more uncertain), there will be certain criteria that determine which studies are preferable to select.

The aim of this study is to describe the criteria used by researchers in the past and develop a list of considerations they should use in the future. We believe it crucial to transparently justify replication target selection to ensure that we do not just replicate *any* study, but have sound reasoning behind our choices. The reviewer flags that the initial list of considerations might be missing some (important) items (i.e., motivation for the replication). We agree it is likely that our initial list misses some considerations that we (the author team) did not think of or agreed on.

We added a clarifying sentence:

- line 129:

The authors acknowledge that they might have missed some crucial considerations when constructing the preliminary list of considerations.

However, we designed our study such that we will hopefully uncover common considerations missing in our preliminary list. For example, we added Stage 2 to specifically ask researchers, who previously conducted replication studies like this reviewer, about their considerations. Common themes will be included in the list of items. Moreover, during the Delphi procedure, participants will have the chance to suggest additional considerations which might still be missing. While we appreciate both the reviewer's insight into the topic and the great suggestion, we perceive this as input regarding the study results (see also [1]) and not the study design, and will thus refrain from adding an item to the preliminary list of considerations.

[5] Let's suppose that your study is now over and the results are in our pocket: what do we do with this checklist? Do you plan to suggest journals to adopt it? Else?

The reviewer raises a great point: we had not been clear on the potential uses and implementations of the proposed tool. First and foremost, this is a tool to be used by authors to transparently AND systematically report their replication target selection process. A great example for transparent selection of a replication target was recently published by Murphy and colleagues (1) (see <https://osf.io/v3wz4/>). While this a useful start to justifying resource allocation, we believe that we could go a step further by asking authors to consider a variety of important aspects when selecting a replication target. If we reach consensus and can build a tool, this could be adopted by authors that are preregistering a replication study or by funding agencies that are providing money for replication studies specifically.

We have made these uses clearer in the manuscript.

- Starting line 88 ff.

To facilitate such a transparent reporting of considerations that led to a replication study, we aim to develop a list of criteria generally regarded as important. *Researchers could use this list to transparently and systematically report their replication target selection process, and metascientists could in turn use these reports to characterise a field's development. A great example for transparent selection of a replication target was recently published by Murphy and colleagues [22]. While this a useful start to justifying resource allocation, we believe that we could go a step further by streamlining this process and offering authors structure and guidance in their selection process. Additionally, a list of considerations would offer a structured tool to funders providing money for replication studies specifically and looking to evaluate the usefulness of a proposal.*

[6] Throughout the paper I implicitly intended "direct replication". But authors do not mention.

This reviewer comment made us aware of an important shortcoming we had not yet considered in our design: the motivation and decision process could potentially differ for direct and conceptual replications. We can take this into account in the first phase of our study by explicitly asking about it, and this rich plurality of the different types of replications and their considerations will surface naturally during the survey and Delphi procedure. A researcher might (1) study certain content and decide to replicate a study from this line of research, or (2) decide to conduct a replication study and identify a suitable target afterwards. The latter bracket will likely consist largely of 'direct replicators' (predominantly meta-scientists). At present, these are just speculations and we have no evidence to back up our line of thinking.

This reviewer comment helped us improve the study design for the stage 2 survey. In our initial design we only asked participants to describe the decision process, enabling us to qualitatively evaluate how the participants came to select the specific replication target (i.e., the replication for which we approached them). We did not include items asking: (1) whether the author self-identified their replication to be direct or conceptual, and (2) what the motivation was for conducting a replication in the first place. Consequently, we added questions asking participants to describe (Q3) and self-identify (Q4) the type of replication they conducted and rephrased Q2 slightly by asking participants to report their motivation. We believe this open-ended way of exploring researchers' motivation will allow for the rich diversity of reasons/motivations people may have had. Moreover, Q3 and Q4 specifically will enable us to contrast responses between direct and conceptual replications and clarify whether the process of selecting a replication target differs between these two groups. Moreover, we will gain additional insights into the motivation behind the replication and be able to explore whether researchers who start out from a well-defined topic of interest more frequently run conceptual replications.

Question 2 now reads:

Q2: Think back on when you chose a replication target. We are interested in how you came to pick the particular study you chose. Please illustrate your *motivation* and decision process to the best of your recollection. [open ended]

Additionally, we added two questions regarding the type of replication.

Q3: Commonly, replication studies fall on a continuum between *direct* and *conceptual* replications. Direct and close replications aim to mirror the original study design as closely as possible. In other cases, replications are extended by adding a new condition to the original design. And in yet other cases, conceptual replications allow for design adjustments to understand boundary conditions of a phenomenon and to build theory. With a replication that you conducted in mind, can you describe what type of replication it was (please give a description for any terms you may use)? [open-ended].

Q4:
If you had to pick one description, which label would fit your replication the best?.

- Close/direct replication
- Conceptual replication
- Other, please specify [text option]

We have also adapted the manuscript to acknowledge this additional aim. We added a clarifying sentence highlighting that we talk about both types of replications.

- Line 17f.

One key element of these efforts involves *various forms* of replication. Close replications aim to mirror the original study as closely as possible, *allowing for example for better estimates and correction of false positives*, whereas conceptual replications *change elements of the original study* to allow for understanding boundary conditions (e.g., by changing measurement and manipulations) and theory building of a phenomenon.

- Starting line 139 ff. :

Additionally, the survey enables insight into the specifics of the selection process and whether it differs depending on the researcher's motivation for conducting a replication and the type of replication. Different considerations might apply to replications that can be more readily termed close replications (e.g., more methodological) than to replications that are more conceptual (e.g., more theoretical). Mapping researcher considerations onto the different types of replications will bring the field one step closer to more explicitly matching outstanding questions for a specific phenomenon with the type of replication that most efficiently answers them (e.g., if a type 1 error is expected, a close replication would be the best match). Although in reality there are many different forms a replication can take (see e.g., 28,29) the distinction between close (direct) and conceptual replication is most common and well-known by researchers, which is why for the current survey we examine the relationship with researcher motivations and these articulated ends of the continuum.

Massimo Grassi

References:

- Klein, R. A., Vianello, M., Hasselman, F., Adams, B. G., Adams Jr, R. B., Alper, S., ... & Batra, R. (2018). Many Labs 2: Investigating variation in replicability across samples and settings. *Advances in Methods and Practices in Psychological Science*, 1(4), 443-490.
- Grassi, M., Crotti, C., Giofrè, D., Boedker, I., & Toffalini, E. (2020). Two replications of Raymond, Shapiro, and Arnell (1992), The Attentional Blink. *Behavior Research Methods*, 1-13.
-

Reviewer: 2

Comments to the Author(s)

Summary of Review:

Kudos to the authors. This was a thought-provoking and enjoyable read. I applaud the pursuit of a registered report publishing format for a project that is descriptive and exploratory. The study is well-designed and the manuscript will make a nice contribution to this interesting, important, and timely literature. None of my comments or concerns or suggestions are “deal makers” or “deal breakers” for me, but I invite the authors to consider editing the study plan and manuscript to address my comments, concerns, and questions as they see fit.

We would like to thank the reviewer for their encouraging words and the insightful comments, which prompted us to make some meaningful changes which improved the present project.

Abstract & Introduction:

[1] I am a bit concerned about the implication that the motivations behind replication work are different from those of any work, novel or replication. From the abstract: “Given limited resources, replication target selection should be well-justified, systematic, and transparently communicated.” Is this only or particularly true of replication work? Not all work? This issue seems really interesting me, and may deserve some attention in the introduction of the paper, if not in the study design itself. **I would be keen to see a direct comparison of these results to “novel” or “non-replication” work.** Are the same 4 factors identified by Isager et al also driving forces for selection of research targets in a scarcity environment? A high degree of similarity seems likely to me.

This is an interesting question also raised by R1(see [3]). We do not wish to argue that it would not be interesting to transparently communicate the reasoning behind original studies. However, we think replications are a special and interesting case. We now include a motivation to consider replication study selection specifically in the manuscript.

- line 73 ff. :

To be clear, we believe that science would benefit from transparently reporting the decisions that led to the genesis of all studies. However, we argue that there is good reason to consider the decision process for replication studies separately from original studies. First, the motivation of and reasoning behind replication studies might differ. While many original studies explore new claims based on theoretical reasoning and previous literature, replication studies have in the past frequently been motivated by the intent to corroborate existing empirical results. Second, the room for a replication to add to a field's knowledge base can be more readily quantified since the primary function of a replication is to reduce uncertainty about existing results (whereas original research can have many different functions, some of which are hard to represent quantitatively). Therefore the selection process may be optimized more

easily for replication studies. Third, due to the lack of being able to play the “novelty card” when justifying the study authors may be facilitated by a systematic approach. With replication and self-correction being deemed important elements of a scientific field [24], a more systematic and transparently documented replication selection process can help characterize - and signal potential points of improvement for - a field’s maturation.

[2] It seems the most common term for psychology’s decade (and counting) of turmoil is “replication crisis,” which ties directly to your paper’s stated mission by using “replication” in the description. I’m interested in the decision not to use that term as you list descriptions of the era of uncertainty in the first sentence of the paper.

There was no reason for us to avoid this term. We added it to the list.

- **Line 3:**

“Researchers have grown increasingly sceptical of the reliability of previously accepted findings, a situation characterized as a crisis of confidence, reproducibility, replication, or credibility. [1,2]”

[3] Is there empirical support/a citation for “a sharp uptick in replication studies”?

There is a reference from 2012, which we initially did not want to include as it might be outdated. To our knowledge, there is no recent one. We included the 2012 reference, but highlight its year of publication. Authors AvtV and PMI are currently working on updating the prevalence estimate of replications in psychology specifically. We might be able to include this reference in the final version of this manuscript.

- **Line 22 ff. :**

A large increase in articles concerned with theoretical and philosophical discussions on replication and replicability is coupled with a sharp uptick in the number of empirical replication studies being conducted [for numbers until May 2012, see 8].

[4] At the end of the second paragraph, perhaps you should unpack what exactly they mean by “did not replicate”?

We specified this in the manuscript.

- **Line 27 f. :**

Only a third of the original studies (36.1%) suggested a statistically significant effect (i.e., $p < .05$) and less than half (41.9%) of the original confidence intervals included the replicated effect size [9].

[5] Generally, I think the intro slightly fails to fully motivate the importance of the study. Is this work prescriptive in any way? What do we hope to learn and what impact will that have on the field?

This concern was similarly raised by R1 (see comment 5). Unfortunately, we were not clear about the potential uses and implementations of the proposed tool, and thus failed to motivate its importance. The tool would be prescriptive in that it offers guidance and structure in what aspects to consider, think about, or evaluate when selecting a replication target. We will refrain from offering a “decision

tree” as we believe that the process of selecting a replication target is too complex to be captured by a single one-size-fits-all procedure. However, the tool will offer insight into this complex process and will allow individual researchers to develop well-justified customized procedures for replication target selection that are transparently reported.

We have made these uses clearer in the manuscript.

- Starting line 88 ff. :

To facilitate such a transparent reporting of considerations that led to a replication study, we aim to develop a list of criteria generally regarded as important. *Researchers could use this list to transparently and systematically report their replication target selection process, and meta-scientists could in turn use these reports to characterise a field’s development. A great example for transparent selection of a replication target was recently published by Murphy and colleagues [22]. While this a useful start to justifying resource allocation, we believe that we could go a step further by streamlining this process and offering authors structure and guidance in their selection process. Additionally, a list of considerations would offer a structured tool to funders providing money for replication studies specifically and looking to evaluate the usefulness of a proposal.*

[6] Sampling:

Social science is a quite broad domain to plan sampling only ~30 participants from. Are you concerned that themes may differ across fields within the social sciences, and that your participants may be overrepresented by one field or underrepresented by others? You plan to ensure a balanced sample in terms of gender, career stage, and country of residence, but field/research area seem much more pertinent, and more likely to bias the themes you identify.

The reviewer raised a valid concern here, and we consequently adapted our sampling plan and Stage 2 questionnaire. Instead of limiting the sample to 20-30 participants, the updated plan proposes to contact all potential participants and have data collection open for a month (with reminders after 1 and 3 weeks). We conducted the literature review to identify potential participants, which resulted in 712 unique email addresses to be contacted. Based on Field et al. (2020) and Alister et al. (2021), we expect a response rate of around 10%.

Changes are reflected in the updated paragraphs concerning the survey participants (starting line 194) and the literature search specifically is described starting line 199. Additionally, we defined the expected response rate in line 246 and following.

Additionally, we now include an item in our Stage 2 survey asking participants to self-identify their field of expertise. The presented subfields are adopted from Aczel et al. (2020) and Alister et al. (2021). Question 1 in the Survey now reads:

Q1: Please choose the psychological field you work in.

- Cognitive and Experimental Psychology
- Clinical and Personality Psychology
- Developmental and Educational Psychology

- Industrial and Organizational Psychology
- Biological and Evolutionary Psychology
- Neuropsychology and Physiological Psychology
- Social Psychology
- Quantitative and Mathematical Psychology
- Human Factors
- Unsure, please specify [text option]
- Other, please specify [text option]

In the Data Analysis section we now specify:

- **line 353**

To explore whether the considerations differ between field of expertise and type of replication, we will stratify the sample and compare subgroups.

Similarly, the initial list for possible delphi participants is made up entirely of what I might call (and consider a decent title for myself as well) “replication evangelists” . Might it be better to include potential delphi experts with more diversity in terms of replication experience and viewpoint?

We thank the reviewer for highlighting this important issue. We agree that more diversity in terms of replication “expertise” is warranted, though we believe that participants need at least some experience with replication target selection. We thus opted to mirror the procedure by Aczel et al. (2020) and now offer participants of the Stage 2 survey the opportunity to sign up for the Delphi procedure. We hope that this will result in a more diverse sample of Delphi experts including both meta-researchers and content-researchers. To reflect the field of psychology, we will additionally stratify content researchers by subfields following the proportions provided by Alistor et al. (2021).

Changes are reflected in...

- **line 367 ff:**

Snowball sampling will be implemented as follows: Prior to registration, we constructed an initial list of 29 potential participants, who we deem knowledgeable in the subjects of replication, replication target selection, methods and statistics, theory, or meta-science. The list can be found in the supplement. *To not only identify “replication experts”, but also content researchers, we offered researchers who participated in the stage 2 survey the option to sign up for the Delphi procedure.*

- **line 399 ff:**

If after 1 month, our sampling procedure resulted in more than 30 participants, we will proceed to the Delphi process, provided that the sample is balanced with regard to gender, career level, and country of residence. *Additionally, we will stratify participants by their research field similar to [34].*

[7] Discussion:

I'd like to see a little more pre-commitment to how conclusions will be reported and discussed. The current manuscript ends rather abruptly and leaves me wondering what the final conclusions will entail.

We added an additional section highlighting the expected data and reporting of results:

- line 452 ff. :

Expected reporting of results

During stage 2, we will produce: quantitative data (i.e., importance ratings), qualitative data (i.e., participants responses and corresponding codes), documents containing reflections and potential sources of bias from coding authors, and an updated list of considerations.

Quantitative data will be summarized using median ratings and IQR and presented in tabular form. We will report identified codes and associated frequencies. Reflections of the coding authors and the updated list of considerations will be made available at OSF. Reflections and reasoning behind what qualified as a theme will be discussed in the manuscript, leading to intermediate conclusions about how psychological researchers select replication targets.

During stage 3, we will produce quantitative (i.e., importance ratings), and qualitative data (elaborations from participants), feedback reports for each round, and a selection of items, which participants agreed upon (i.e., the final checklist), and potentially items that no consensus was reached for. Median ratings and IQR for each item across the rounds will be presented in a table. We will report our definitive checklist, highlighting in particular the items that reached consensus, but also those that did not. Feedback reports will be uploaded to OSF, summarizing also the qualitative input from the Delphi process. These results will allow us to discuss and suggest relevant considerations for future researchers, discuss implications for psychological science, and potentially other social sciences and signal potential direction for future research.

Recommended Decision:

Again, I leave it to the authors to decide if they should change their study plan or edit their manuscript in response to my review. This project will make a contribution, and I look forward to reading future drafts of the manuscript. I recommend an in principle acceptance following minor revisions.

Reviewer: 3

Comments to the Author(s)

1. The scientific validity of the research question(s).

• Concise rationale that makes an excellent case for this study. I wish the researchers the best of luck with this study.

Thank you very much, we appreciate the kind words.

2. The logic, rationale, and plausibility of the proposed hypotheses.

[1] Not applicable although (following the Mixed Methods Article Reporting

Standards; <https://apastyle.apa.org/jars/mixed-table-1.pdf>) can you specify your aim or aims for this study?

We have clarified our aim:

- line 101 ff.:

The present study

We argue that the involvement of the wider scientific community is crucial *when designing a list of considerations to be used for transparent and systematic replication target selection. In this project, we aim to (1) describe the considerations generally regarded as important by psychological researchers and (2) construct a list of considerations to be consulted when selecting future replication studies in psychology.* To ensure that our results reflect considerations of the selection process that are generally regarded as important by the *psychological* community, we will employ a consensus-based method. More precisely, we will use a Delphi approach to expound the considerations and criteria researchers commonly deem important when selecting a replication target.

3. The soundness and feasibility of the methodology and analysis pipeline (including statistical power analysis where applicable).

[2] Following the Mixed Methods Article Reporting Standards, what is the study design? Is this an exploratory sequential design?

We thank the reviewer for alerting us to make our design more explicit, as this is a practice that we wholeheartedly encourage. We think that the definition of an exploratory sequential design fits and have specified this in the manuscript. We additionally specified what type of data will be collected and how.

- Line 111 ff. :

The Delphi process, *which has the goal of developing consensus on a given topic or issue, is one of the most frequently used methods across multiple fields [26]. The Delphi process, as applied in this setting, is descriptive and can be considered an exploratory sequential mixed*

methods design. It is an iterative process, in which judgements from 'informed individuals' are collected in the form of questionnaire responses. ***The questionnaire collects both quantitative data in the form of importance ratings and qualitative data in form of suggestions and opinions on judgements.*** Over several rounds, consensus on several judgements or opinions emerges [27]. We have chosen this method for use in the current project, as it allows for including researchers from all over the globe, ensures anonymity of responses which allows participants to disagree more freely [26], and is most likely to yield results which reflect the opinions of the group as a whole, rather than capturing the views of a select few outspoken individuals.

[3] Would it also be useful to identify your population of interest/potential audience? Are you interested in all sciences or just specific disciplines (e.g. Psychology and/or cognition sciences) - the size of your population of interest should then feature in sampling and recruitment. For example, if you focused on cognition sciences, you might want to ensure you have representation across the difference components of these sciences. Or if you focus on psychology, you might want to perhaps seek representation across the range of psychological approaches. And perhaps you would only focus on disciplines/sub-disciplines in which replication is arguably of interest (e.g. Experimental approaches).

We have refined our population of interest for the survey to psychological researchers and adapted our search strategy accordingly (also reflected by multiple small changes in the manuscript). In line with this, we refined our Web of Science search to: Psychology Biological, Psychology, Psychology Multidisciplinary, Psychology Applied, Psychology Clinical, Psychology Social, Psychology Educational, Psychology Experimental, Psychology Developmental, Behavioral Sciences, and Psychology Mathematical. These categories are adapted from Field at al. (2020) and Alistor et al (2020). While the Web of Science categories can be arbitrary, we believe that this selection maps closely onto the target population (see also Q1 in response to R2 [6]). Moreover, the sampling plan for the survey was refined. We now plan to contact authors of all replication studies in psychology over published/registered in the past five years. While self-selection bias might still bias the composition of the final sample, we believe that our effort should lead to the sample being relatively representative of psychological researchers conducting replication studies.

Changes are reflected throughout the manuscript. Please see the paragraphs "Participants of the Survey" starting line 195.

• Stage 2

[4] Note that you will 'evaluate whether saturation has been reached after 20-30 responses'. This is quite a range. Saturation is a contentious concept, so can you perhaps locate your use of it in the literature and clarify how it will be operationalised in this study (see e.g., doi.org/10.1080/2159676X.2019.1704846).

Based on the reviewer's suggestions to increase sample size (see reviewer 2 [6]), we revised our sampling procedure and sample size determination for the Stage 2 survey and removed the concept of saturation from the manuscript.

We initially planned to sample participants incrementally. Starting after 20 responses, we planned to determine whether saturation had been reached in incremental steps of 3. Based on previous research, we expected to reach saturation after around 30 responses. In this scenario saturation would have determined when to stop collecting data. Now, we base the proposed sample size on the available participant pool identified through a systematic search. We systematically searched Web of Science and the OSF registries to identify authors, who registered or published a replication study in the past five years. We identified a participant pool of 712 candidates. Instead of sampling incrementally, we will contact *all* candidates for the survey. Data collection will be open for a month with reminder emails after one and three weeks. We anticipate a response rate of around 10% and will analyze all data available after a month. We believe that using this sampling strategy will result in a sufficiently large data set including both qualitative and quantitative data.

Our sample size determination now reads:

- Line 222ff:

The survey in stage 2 serves as a pilot to inform the list of items provided for the first round of the Delphi process. Sample size determination for qualitative work is complex and depends on a variety of factors such as the scope and nature of the research, the quality of the data collected, and what resources are available. *Here, we based sample size considerations on the available pool of potential participants.* Typically, qualitative studies report between 20-30 participants. *For the purpose of our project, we deem it crucial for our sample to be large enough to be reflective of the consensus in the field. We identified a total of 712 potential participants meaning that with an expected response rate of 10%, our sample will be roughly twice as large as recommended. However, if the response rate is lower than expected resulting in fewer than 30 participants, we will extract additional candidates from the past 5-10 years and repeat sampling.*

[5] Can you clarify the tense of 'choose' in Stage 2. Are you asking participants to report on their decision making (so, 'chosen', past tense)? Or are you asking for their opinion and/or attitude about ideal decision making for current (projects they are currently working on) and/or future replications? Note that at the start, you focus on people who have 'conducted' (past tense) a replication but later mention that you will include people who are conducting (so potentially present tense as they may be currently working on replication decision, although they may have already made the decision (so, past tense). I think it is important to clarify if you are looking for self-report of decision making or opinions/attitudes about ideal decision making.

This is an important distinction, and we appreciate that the reviewer picked up on this! We have made several small changes to clarify this issue. We clarified that we are concerned with people who underwent the decision process of selecting a replication target, regardless of whether this replication was conducted or only registered.

- Line 131:

To overcome this, we will include an additional survey round with individuals who selected a replication target in the past.

- Line 195:

We plan to sample psychological researchers who have previously selected a replication target, identified as having either conducted or registered a replication study.

We have additionally clarified that we are interested in the self-report of decision making process of selecting a replication target:

- **Line 251:**

To gain insight into the replication target selection process and pilot the preliminary list of items, we constructed an online survey with eleven questions. The aim of the survey is to (1) pilot the considerations included in the preliminary list and those the author group was undecided about, and (2) to capture considerations not mentioned in the preliminary list. The former will be achieved through closed questions rated on Likert scales, and the latter through open questions leaving room for suggestions and additional information.

- **Line 266:**

To gain insight into how replication targets are selected in practice, we will ask our participants to illustrate *what motivated them to replicate* and how they came to pick the particular replication target they chose (open question Q2).

[6] How do the Likert scale fit with qualitative survey design? How will they feature in the analysis?

The survey includes both open and closed questions, so it is not completely qualitative. The aim of the study was to (1) pilot the considerations we collected in the preliminary list and those the author group was undecided about, and (2) to capture considerations we missed when constructing the preliminary list. The former is achieved through closed questions rated on Likert scales, and the latter through open questions, leaving room for suggestions and additional information.

We have made changes to the manuscript to explain this better:

- **Line 251:**

To gain insight into the replication target selection process and pilot the preliminary list of items, we constructed an online survey with eleven questions. The aim of the survey is to (1) pilot the considerations included in the preliminary list and those the author group was undecided about, and (2) to capture considerations not mentioned in the preliminary list. The former will be achieved through closed questions rated on Likert scales, and the latter through open questions leaving room for suggestions and additional information.

The data analysis of the closed questions is explained in the Data Analysis section.

- **Line 350:**

Closed-ended items will be evaluated using the median rating and interquartile range (IQR), a measure of dispersion around the median capturing the middle 50% of observations [36]. Old items with a median rating of 3 and an IQR of 2 or lower will be excluded from the list, and new items with a median rating of 7 and an IQR of 2 or lower will be included. *To explore whether the considerations differ between field of expertise and type of replication, we will stratify the sample and compare subgroups.*

[7] As this is a qualitative component (although if the Likert scales are key then it will be mixed methods), can you follow the Journal Article Reporting Standards for Qualitative Methods (<https://apastyle.apa.org/jars/qual-table-1.pdf>) and include the researcher description and the researcher-participant relationship.

Thank you for this valuable suggestion. We have added researcher descriptions.

- line 162 ff:

MMP has previously published work on replication target selection in clinical psychology [20]. Her interest in the topic stems from a background in clinical psychology and the realization that sometimes "shaky" effects are translated into clinical practice. In her opinion, (1) treatments should be recommended only with sufficient evidence, also achieved by replications, and (2) which studies to replicate and how should be determined by evaluating a set of candidate studies. DvR has published theoretical work on replications [19,20,30] and has conducted empirical replications [31,32]. SMF has also published theoretical and empirical works concerning replication [19,31]. Frustration with (sometimes) inefficient use of resources and insufficiently justified reasoning behind conducting replications drives her interest in providing researchers with the means to help systematize the replication target selection process, which can be difficult to navigate. PMI has previously authored theoretical work on replication target selection [17,21]. AvtV has previously published about theoretical and practical aspects of conducting replications [33], has conducted large scale and Registered Replication projects [9,34–36], and is involved in theoretical and meta scientific work on replication target selection [17]. Her experience in analysing replications within psychology strengthen her belief that more explicit characterising of (the process of conducting) replications and their various functions can be a step towards making replication common in research lifecycles and towards theory building through conducting progressive types of replications.

Additionally, we added descriptions about the anticipated researcher-participant relationship. This will be updated after data-collection.

- line 236 ff. :

Some of the participants might be distant colleagues of the research team. However, the authors will not interact with participants as data is collected anonymously online. Nonetheless, the author names are disclosed during the survey, which might impact data collection.

- line 389 ff.:

We anticipate the sample to consist (mostly) of researchers who are distant colleagues or perhaps one-time collaborators with some of the author team. Our contact with them in the context of the study will be distant.

- Stage 3

[8] Question about sample size; this seems based purely on stats rather than an underpinning theory of consensus. Surely a Delphi is akin to an election, so that any consensus is dependent on the

number/proportion of the people from the relevant communities engaged in it. This seems to be a relatively unobtrusive study that should be easy to conduct at a much larger scale. If you compare this sample size to the number involved in the Open Science Collaboration, it pales in comparison.

While a Delphi procedure is unobtrusive it is time-consuming. Participants need to go through several rounds of providing both quantitative and qualitative feedback and receiving feedback reports from the research team. As such, the Delphi procedure differs from an election and asks for more involvement from the side of the participant. We opted for a sample size, which based on empirical evidence, should result in stable consensus, while not asking too much of participants. Based on the reviewer comment however, we are now careful to state that the anticipated 30 participants are the minimum sample size we believe necessary to reach stable consensus. We welcome a larger sample and now include a second line of participant sample (i.e., offering Survey participants to take part in the Delphi survey too) to increase our sample size.

Changes are reflected:

[9] Great to see you offering co-authorship for your Delphi participants. I am currently doing this (offering participants the opportunity to opt-in to co-authorship) on a study. Is it worth adding that you consider them to be 'investigators' (in semantic terms, participants are collectively helping you/each other to investigate this topic) in relation to CRediT authorship taxonomy? It might be worth clarifying if co-authorship is opt-in or a requirement of participation.

This is a great suggestion. We added this to our manuscript.

- **Line 386:**

Authorship is voluntary and not a prerequisite for participation. If Delphi experts decide to identify as authors they will be considered investigators according to the CRediT taxonomy.

[10] Personally, I think you could be much more ambitious about your sample size.

Thank you for this suggestion. This concern was also raised by reviewer 2 (point 6). In response, we now removed the limitations for sample sizes of both the Stage 2 survey and Stage 3 Delphi procedure.

Changes are reflected for Stage 2 starting line 240 and for Stage 3 in line 397.

4. Whether the clarity and degree of methodological detail would be sufficient to replicate the proposed experimental procedures and analysis pipeline.

- As a consensus study, replicability is not an issue. That is, repeating this study in the future should hopefully lead to different findings. If the findings were replicated, that would unfortunately signal that open science has remained stagnant.

5. Whether the authors provide a sufficiently clear and detailed description of the methods to prevent undisclosed flexibility in the experimental procedures or analysis pipeline.

- There is disclosed flexibility in Stage 2 that is consistent with qualitative analysis.

[11] There is undisclosed flexibility in how the researchers will determine if saturation has been reached in the data analysis. I would recommend clarifying how saturation will be operationalised.

As we also say in our response to point [4], the sampling plan was adapted. The sample size is now motivated by the available participant pool and we will analyze all data available.

[12] There is flexibility in the rounds of a Delphi study. That is, the researchers may decide to exclude some items and/or add new items. Can you perhaps add a section detailing how analysis will be conducted between each round? What will you do, for example, with missing data? Given the importance of flexibility, perhaps it would be useful to also add how you will document the decisions between each round (e.g. transparency log, sharing analysis code used at each round).

We added a Data Analysis Plan section to explain the data analysis process in Stage 3.

- **Line 431ff:**

Data Analysis Plan

Data analysis will be performed after each Delphi round. Quantitative items will be analyzed using medians and IQR and the distribution of ratings will be visualized using histograms. Items with a median rating of 6 or more and IQR of 2 or less will be included in the final list of considerations. Items with a median rating lower than 6 and an IQR of 2 or less will be excluded. Qualitative responses will be summarized by MMP and discussed by the author group. We will count how many individuals mentioned a certain concern or suggestion. The list items will be revised based on frequently mentioned suggestions. We consciously refrain from defining frequently a priori to allow us to flexibly respond to concerns and suggestions later on. We anticipate no incomplete data reports as we will force participants to answer every item. If participants have no suggestions, they will be instructed to answer open questions with “none”. If due to attrition, participants do not join subsequent Delphi rounds, we proceed with the remaining experts.

After each round of data analysis, MMP will construct a structured feedback report for the participants. Items for which consensus was reached will not be included in the summary report to the participants. In the feedback report we will: (1) reply to frequently raised general concerns if there are any, and (2) present items for which no consensus was reached. For each item we will present the histogram of responses, highlight revisions if necessary, and address item-specific concerns. Summary reports and the invitation for the next round will be sent to participants who responded to the previous round.

6. Whether the authors have considered sufficient outcome-neutral conditions (e.g. absence of floor or ceiling effects; positive controls; other quality checks) for ensuring that the results obtained are able to test the stated hypotheses.

- Not applicable for this design.

Appendix B

Merle-Marie Pittelkow
Department of Psychology,
Faculty of Behavioral and Social Sciences,
University of Groningen
+31 50 36 32699
m.pittelkow@rug.nl

Dr. Chris Chambers

Subject Editor Registered Reports, *RSOS*

On behalf of my co-authors, I am resubmitting our Registered Report “The Process of Replication Target Selection: What to Consider?” manuscript ID RSOS-210586.R1.

We have followed our proposed methods as closely as possible, with the exception of a few deviations, which were communicated to the editorial team. All deviations are clearly highlighted in the final manuscript. We did not alter the introduction but made some minor changes to the method section to change the tense from future to past. For your convenience, we submitted one manuscript with and one without changes highlighted. No data was collected prior to the date of IPA and the timepoint(s) of data-collection are communicated in our manuscript.

We uploaded our materials, data, analysis files, and the approved stage 1 protocol to OSF (https://osf.io/j7ksu/?view_only=) and provide the link in a designated section on Page 13.

Also on behalf of my co-authors, I would like to thank you again for giving us the opportunity to run this project as a Registered Report. This was a great experience and we think we learned a lot during the process.

Sincerely,

Merle-Marie Pittelkow

Appendix C

Associate Editor Comments to Author (Professor Chris Chambers):

Associate Editor: 1

Comments to the Author:

I have now obtained evaluations of your submission from two of the reviewers who assessed your manuscript at Stage 1. Both are positive, as is my own assessment. The reviews nevertheless contain some helpful reflections and recommendations for minor improvements. Concerning the suggestion by Reviewer 1 to shorten the manuscript, and I felt the length was acceptable as the manuscript is well organised and it doesn't feel overly long. However, I am happy for you to shorten it if you prefer. If you do so, please focus any such reduction on results and discussion; please do not remove any content that was approved at Stage 1.

Concerning Reviewer 3's comment about using the OSF, this is an acceptable repository from the journal's point of view.

I look forward to receiving your revised submission and response to the reviewers, after which I expect to be able to issue full acceptance without further in-depth review.

Thank you for your kind words and thank you for giving us the opportunity to conduct this exploratory, partly qualitative project as a Registered Report in the first place. The process has been highly educational for me and I am sure I will submit future projects as RRs again.

The reviewers made some great suggestions for improvement and we believe to have responded to them fully. We opted against shortening the manuscript. While we understand that the result section (and especially the qualitative results) might be longer than what is common for consensus-based projects, we believe that shortening this section would distort our results.

Additionally, we noticed that our use of "supplementary material" might have caused some confusion. We therefore now clearly refer to either the supplement or the additional files provided on OSF instead of referring to both as supplementary files.

We hope that the changes we have made will enable you to issue full acceptance.

Kind regards,

Merle-Marie Pittelkow (also on behalf of my co-authors)

Comments to Author:

Reviewer: 1

Comments to the Author(s)

I read the stage 2 and my opinion oscillates between "accept as is" and "minor revisions". The main problem I see now in the paper is length. Although the final paper is super transparent and reports everything, I think it would be more important for the community if it was shorter. My 2

cents suggestion is to move part of it (ie the process) somewhere else and keep beginning and end.

Minors.

1. I saw that in the "The present study" section there are still a few future tense. I saw authors removed them from the rest of the paper. To me, that part looks it should be written in past tense. But I'm not an English speaker therefore please drop this point no sense.

Thank you for pointing this out. We have updated the section accordingly.

2. I found captions a bit too short and with few details. In such a long paper, figure, tables (and captions) are vital, I think. Figure 3 in particular is there with no much explanation.

We have updated the captions for all figures and tables mentioned in the result section to make them more descriptive. We refrained from updating captions that were already presented in the Stage 1 manuscript. Changes can be found on page 14, 15, 16, and 24.

Very good job.

Massimo Grassi

Reviewer: 3

Comments to the Author(s)

This is my second time reviewing a Stage 2 submission, so I am being guided (and will use subheadings to emphasise) the five key issues to consider Royal Society Open Science, Registered Report, 'Reviewer Guidelines' (<https://royalsocietypublishing.org/rsos/registered-reports>). Overall, the paper is easy to read and provides sufficient detail to understand what they did and why, and how they came to the conclusions in creating a transparent checklist.

1) Whether the data are able to test the authors' proposed hypotheses by passing the approved outcome-neutral criteria (such as absence of floor and ceiling effects or success of positive controls or other quality checks). Failure to pass these conditions may lead to manuscript rejection.

Yes. The data are in a folder called 'supplementary files' on the OSF project page and there is a sub folder for all three stages of the Delphi Exercise. In Stage 2, for example, the survey responses are complemented by the qualitative codebook and the reflexive notes. This really helps show the process of the of data analysis. In the manuscript, is it worth specifying what license you have given the data, so that readers can be clear about what they can, and cannot, do with it?

We have added the requested specification in the manuscript (page 13 line 447ff).

In looking at the data, I am wondering if it meets the FAIR principles for data stewardship (see Step 3 in DOI: 10.31234/osf.io/ahdcu). I wonder if using the term 'supplementary file' might limit 'accessibility' (rather than say, the 'data' component in OSF). My understanding is that 'supplementary' is in reference to this manuscript but the OSF project page is for a study, so this reference point might be unclear. When I looked at the project page, why first question was 'where is the data', and it took me some time to figure it out.

We have renamed the OSF folder to “Additional files (materials, data, analysis code)” to be more descriptive. In addition, the README_OSF.pdf file specifies the location of all files and clearly demarcates the data component.

Also, if you add the license, you'll improve the 'reusability' (and a data component can have a different license to the project folder).

Thank you for pointing this out. We have added a CC-By Attribution 4.0 International license for all components of the project folder.

Could you also include copies of the information and consent participants consented to - consent it something I've found is incredibly important in interviewing qualitative researchers about open data (DOI:10/gf429z)? For anyone wanting to look at the data, they may need to see precisely what participants consented to before deciding if they can or should use it.

The informed consent files are already included in the supplementary material (p. 13 and p 46). To link the informed consent files to the collected data we have additionally uploaded them in the specific folders on OSF.

I also wonder if the OSF is the best home for research data. Will this data, for example, be 'discoverable' for researchers using Internet search? Would a specialist data archive, such as the UK Data Service or the Finnish Social Science Data Archive, be a better home or 'steward' for your data, particularly over the long-term?

In reference to the journal as well as the paper the reviewer mentioned earlier (DOI: 10.31234/osf.io/ahdcu), we believe that OSF is an appropriate repository for the data.

Last, the paper embeds URL to relevant places in the OSF project page, although I would recommend following the journal style and citing them. If the urls ever become broken, the additional information in the citation should help readers find it.

In addition to the URL we have also added a DOI, which is linked directly to the OSF project page. This way, the OSF page should stay discoverable even if the url ever gets broken (page 13 line 448)

2) Whether the Introduction, rationale and stated hypotheses are the same as the approved Stage1 submission (required)

Yes, as far as I can tell, they are the same.

3) Whether the authors adhered precisely to the registered experimental procedures

There were some deviations, which are described and justified. In relation the qualitative analysis, the explanation of the deviation is helpful in clarifying the role of the researchers in interpreting the data. I also think explanation of these deviations helps explain the decision in the Discussion section for a checklist for transparent reporting.

4) Where applicable, whether any unregistered exploratory statistical analyses are justified, methodologically sound, and informative.

Not applicable.

5) Whether the authors' conclusions are justified given the data. Please note that editorial decisions will not be based on the perceived importance, novelty, or conclusiveness of the results.

Yes. I look forward to seeing the use of this checklist in future research. For the avoidance of any doubt, can you include the final checklist/a reference to the final checklist. E.g. "The final checklist (reference)..." p. 24, first line/line 892.

Thank you for this suggestion. We have now added the final checklist as an appendix to the paper (page 34ff), which is mentioned on page 24 line 869) and included the url to the downloadable version of the checklist on OSF.

Typographical

Is the citation on page four, first line, correct "a consensus-based method¹". Looking at the title of the paper cited, it doesn't look relevant.

The "1" in this case marks the first footnote, which is relevant to this sentence. We did not make any changes here.

Peter Branney